# Efficient Management and Scheduling of Massive Remote Sensing Image Datasets

Jiankun Zhu [1], Zhen Zhang [1,*] , Fei Zhao [2], Haoran Su [3], Zhengnan Gu [1] and Leilei Wang [1]

1 School of Geomatics, Anhui University of Science and Technology, Huainan 232001, China
2 China Satellite Communications Co., Ltd., Beijing 100190, China
3 School of Architecture, Harbin Institute of Technology (Shenzhen), Shenzhen 518055, China
* Correspondence: zhangzhen@aust.edu.cn

**Abstract:** The rapid development of remote sensing image sensor technology has led to exponential increases in available image data. The real-time scheduling of gigabyte-level images and the storage and management of massive image datasets are incredibly challenging for current hardware, networking and storage systems. This paper's three novel strategies (ring caching, multi-threading and tile-prefetching mechanisms) are designed to comprehensively optimize the remote sensing image scheduling process from image retrieval, transmission and visualization perspectives. A novel remote sensing image management and scheduling system (RSIMSS) is designed using these three strategies as its core algorithm, the PostgreSQL database and HDFS distributed file system as its underlying storage system, and the multilayer Hilbert spatial index and image tile pyramid to organize massive remote sensing image datasets. Test results show that the RSIMSS provides efficient and stable image storage performance and allows real-time image scheduling and view roaming.

**Keywords:** remote sensing; distributed storage system; big data; scheduling optimization

## 1. Introduction

The development of spatial information systems and demand for remote sensing data have led to the rapid development of image sensor technology. Remote sensing data resources are gradually being enriched and are widely used in military, disaster prevention, environmental protection, earth navigation and other applications. These trends have created challenges for current geographic information disciplines. On the one hand, the large number of earth observation satellites means an increasingly apparent multi-source heterogeneity among remote sensing images [1], while the scale of image data are gradually increasing [2]. Managing massive amounts of multi-source heterogeneous remote sensing image data poses extensive challenges to storage systems regarding capacity, performance and cost [3]. On the other hand, the development of sensor technology has improved image resolution, with the file size of a single image reaching the gigabyte level. Due to networking and hardware technology limitations, it is difficult to achieve real-time reading and visualization of such large amounts of data [4]. Slow data loading will affect the efficiency of image use. Therefore, remote sensing databases urgently require improved management and scheduling methods to adapt to current needs.

Currently, the storage architecture types of massive remote sensing image databases can be divided into centralized server clusters, such as the Terra Server [5] and China Resources Satellite Application Center [6], and distributed server clusters, such as EOS-DIS [7], Google Earth [8] and Bing Maps [9]. Centralized storage has a simple deployment structure, convenient post-maintenance requirements and high reliability. However, due to the architecture design, there are inherent deficiencies in horizontal expansion, disaster recovery mechanisms and fault recovery. Distributed storage has obvious advantages in cost, compatibility and scalability. With the increasing amounts of data, its benefits are

becoming more obvious. It is one of the most promising potential solutions for storing massive amounts of remote sensing image data. Currently, the mainstream distributed file systems mainly include HDFS [10], Lustre [11], FastDFS [12], GridFS [13], MooseFS [14], GlusterFS [15] and CEPH [16]. FastDFS, GridFS and GlusterFS are suitable for file-based online services, such as video and still images. Lustre requires the support of special devices typically used in high-performance computing. Remote sensing image files are very large—a single file can reach the gigabyte size—and once stored without modification, in line with the design concept of HDFS and MooseFS [17]. However, MooseFS is typically applied to single-cluster deployments. As the cluster scale expands, it is prone to uneven loads, and a greater risk of instability [18]. However, distributed file systems still have some limitations in storing and managing massive remote sensing images. Metadata-based attribute retrieval and location-based spatial retrieval are issues that must be considered when managing remote sensing images. Although distributed file systems can efficiently store massive remote sensing images, their retrieval performance is insufficient to support the complex retrieval conditions necessary for multi-source heterogeneous remote sensing images.

Remote sensing image scheduling is an interdisciplinary research field which scholars in geographic information and computer science have ignored. Therefore, there are few or one-sided studies in this field. The two most commonly used methods to improve the efficiency of remote sensing image data scheduling are spatial indexes [19–22] and tile pyramids [23–25]. In a spatial index, the two-dimensional plane is reduced to one-dimensional coding by an index algorithm. The complex spatial intersection operation is converted into simple coding matching, shortening the image retrieval time. Tile pyramid technology achieves rapid image retrieval by slicing remote sensing images and generating multi-resolution images. The amount of data needed to be processed for single-image retrieval and display is reduced. Presently, research on spatial index algorithms and tile pyramid technology is relatively mature and has been applied to many remote sensing image management software systems. Both technologies optimize remote sensing image data organization, greatly enhancing image scheduling performance. However, the image data scheduling process still involves multiple stages, such as data retrieval, transmission, and image visualization. This can result in data loading stalls and a lack of real-time response.

Considering the shortcomings and limitations of the current research on remote sensing image management and scheduling, this paper designs and implements a new remote sensing image management and scheduling system (RSIMSS) which realizes the efficient management and real-time scheduling of massive remote sensing image data. Its characteristics and advantages are:

(1) Based on the spatial index algorithm and tile pyramid model research, the scheduling process of remote sensing images is deeply analyzed regarding computer network transmission, and three new scheduling mechanisms (ring caching, multithreading and tile-prefetching mechanism) are designed to optimize the scheduling process. As the three mechanisms work together, the remote sensing image data scheduling achieves second-level real-time response rates.

(2) According to the spatial distribution characteristics of massive multi-source heterogeneous remote sensing image datasets, a spatial index based on a multi-layer Hilbert grid is constructed to achieve efficient retrieval. The PostgreSQL database cluster is distributed with the same Hilbert grid, and the multithreading mechanism is relied on in the cluster to substantially improve the efficiency of cross-server and cross-database image retrieval.

(3) The distributed file system and relational data library are used to manage remote sensing image data in a hybrid manner. HDFS with high I/O performance and high-capacity scalability is used to store massive unstructured file data. The PostgreSQL relational database with powerful retrieval performance manages structured metadata. RSIMSS takes full advantage of the respective benefits of distributed file systems and relational databases.

The rest of this paper is organized as follows. Section 2 introduces the RSIMSS in detail, including its system architecture, data organization, data scheduling and scheduling optimization strategy. Section 3 tests the storage performance, retrieval performance and scheduling performance of RSIMSS, and discusses the results. Section 4 summarizes the main work of this paper.

## 2. Materials and Methods

### 2.1. The RSIMSS System Architecture

The RSIMSS adopts a modular layered architecture, divided into a data layer, service layer and user layer (Figure 1). The data layer adopts a PostgreSQL database cluster to store the structured metadata of remote sensing images. It constructs a spatial index using a multi-layer Hilbert grid algorithm to improve the efficiency of image retrieval. An HDFS file system is used to store unstructured data of the remote sensing images, which has high I/O performance and good capacity scalability. The service layer is the core of the whole system and achieves image scheduling. It is divided into three modules for data retrieval, transmission, and visualization. The efficiency of image data scheduling is optimized by multi-threading, caching and prefetching mechanisms. The user layer hides the complex internal structure of the entire system and only provides users with a functional interface for data retrieval, preview and download. In addition, the system also reserves some interfaces developed by Java, which is convenient for future expansion of the system.

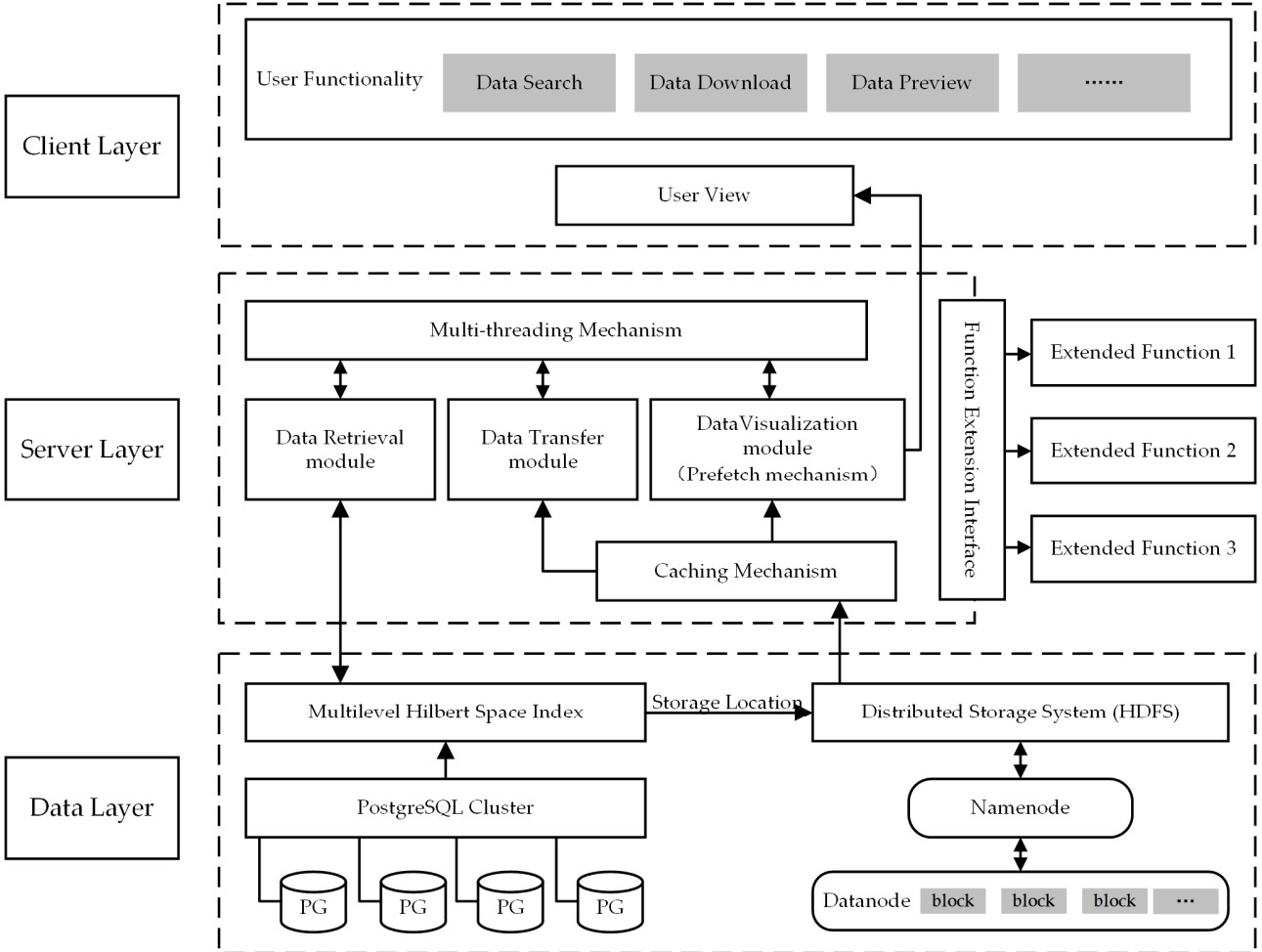

**Figure 1.** The RSIMSS system architecture.

### 2.2. Organization and Storage of Remote Sensing Images

### 2.2.1. Spatial Index Based on a Multilayer Hilbert Grid

The retrieval process must often consider the image's spatial location, access date, resolution, sensors and other multi-attribute information. In addition to the location, other attributes can be expressed and stored in the database as fundamental data types, such as integer, float, string and time. Almost all databases have mature retrieval algorithms for these basic data types. Location and coverage are essential features that distinguish remote sensing images from other data types. Quickly and accurately finding images for a specific geographic space is one of the critical issues in remote sensing data organization and management.

Establishing a spatial index is an effective way to improve the performance of remote sensing image retrieval. Such images represent Earth in three-dimensional space, while the index is a one-dimensional coding. Therefore, the Earth must initially be projected from three-dimensional space into a two-dimensional plane, and the projection planes are then divided into grids. Finally, the grids are encoded to map the two-dimensional projection into a one-dimensional index coding. The Plate Carree projection method is used in this paper. The lower-left corner is the projection coordinate point of the earth's south pole (90° S, 180° W), while the upper-right corner coordinate is that of the north pole (90° N, 180° E). A rectangular projection range is more conducive to grid division (Figure 2).

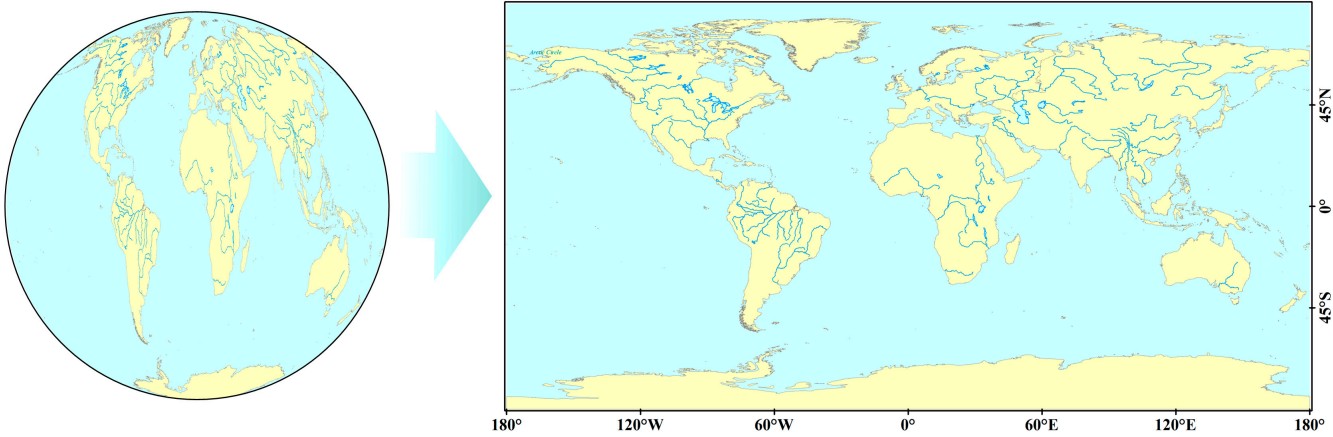

**Figure 2.** Plate Carree projection.

The coding algorithm used in this paper is the Hilbert space-filling curve algorithm. The space-filling curve was first proposed by the mathematician Peano in 1890 [26]. It is essentially a spatial dimension-reduction method. The basic idea is to use a continuous curve to pass through each space unit in turn, only once. As one of the most representative space-filling curves, the Hilbert curve has excellent spatial aggregation and low loss of spatial characteristics. It can better retain local points in multi-dimensional space [27] and is more suitable for remote sensing image retrieval in continuous space (Figure 3).

As remote sensing images from different sources have different resolutions and coverage, a single-layer spatial grid cannot retrieve image data from all potential sources. Therefore, the global plane map is divided into multi-layer grids based on the Hilbert space-filling curve, then it builds a multi-layer grid spatial index on this basis. The first-level grid divides the world into four equal parts, with each subsequent grid dividing the previous one into four equal parts. The size of the k-level grid is (360°/2 k, 180°/2 k). All grid levels are encoded according to the direction of the Hilbert curve with a combination of numbers from 1 to 4. Each level of grid coding is extended based on the previous level of grid coding. Assuming that the level 1 grid coding is C1, the k-level grid coding is C1C2C2 . . . Ck (Figure 4). Finally, the grid is iteratively calculated based on the remote sensing image's central coordinates and coverage area. The iteration starts with the 1-level grid and iteratively divides the grid where the image center is located. The criterion for terminating

an iteration is that the area of the image is greater than the grid area. The grid encoding of the final iteration is used as the spatial index of the remote sensing image.

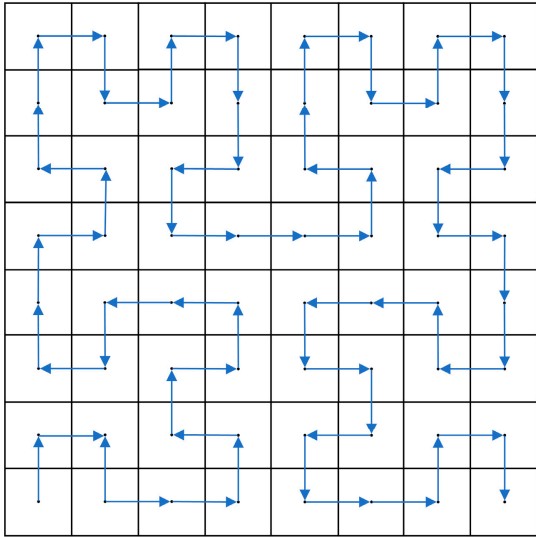

**Figure 3.** Hilbert curve.

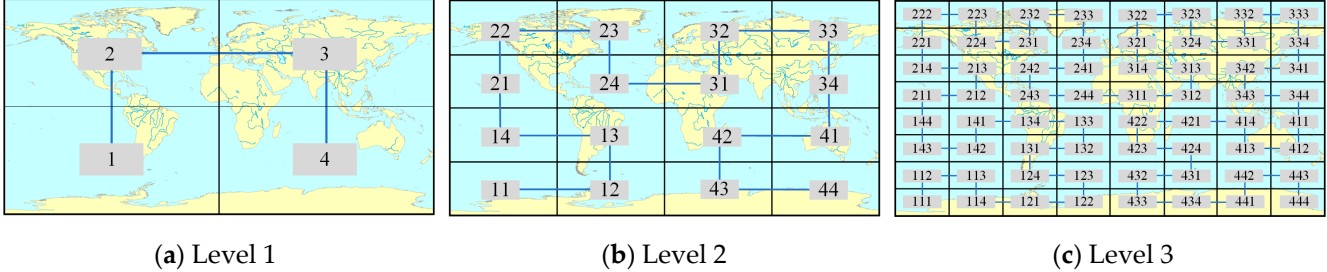

(**a**) Level 1          (**b**) Level 2          (**c**) Level 3

**Figure 4.** Multilevel Hilbert grid coding.

2.2.2. Structural Design and Automatic Analysis of Remote Sensing Image Metadata

In this study, the FGDC Metadata Standard [28] and GB/T 35643-2017 Remote Sensing Image Metadata Standard [29] were used as references to set a unified metadata standard for multi-source heterogeneous remote sensing image data. The metadata were divided into 'Product Set', 'Image Public Property' and 'Image Specific Property'. The 'Product Set' is used to store the attribute information of an image dataset from a particular data source that is processed by the same method. The primary purpose of its design is to speed up the retrieval efficiency of images from a particular data source. Due to the different formats of remote sensing image metadata from various sources, other metadata (except product set information) are divided into two parts: 'Image Public Property' and 'Image Specific Property'. This ensures the integrity of the remote sensing image metadata and improves the image retrieval performance.

Remote sensing image metadata are structured data, so the RSIMSS deploys a PostgreSQL database cluster to realize the storage of multi-source heterogeneous remote sensing image metadata. PostgreSQL is a very popular open-source relational database, and its extension module PostGIS allows it to support many geographic functions. The distributed database system has more efficient data management and retrieval performance than other databases [30]. Part of the 'Product Set' table structure design is shown in Table 1, while part of the 'Image Public Property' is in Table 2. As the 'Image Specific Property' tables of remote sensing images from different data sources have different structures, this article will not include separate detailed descriptions.

**Table 1.** 'Product Set' table structure.

| Field Name | Data Type | Field Description |
|:---:|:---:|:---:|
| Id | integer | Unique identification of the product set |
| Name | text | Name of product set |
| Satellite | text | Satellite parameter information |
| Sensor | text | Sensor parameter information |
| Institution | text | Institution |
| Type | text | Type of product set |
| Gridlevel | integer | Corresponding multilayer grid levels calculated by image size |
| Area | double | Average area covered by remote sensing imagery ($km^2$) |
| Interval | double | Period of sampling at the same position |
| Resolution | double | Resolution of remote sensing images |
| Stime | timestamp | Time taken to generate the first image of the product set |
| Etime | timestamp | Time taken to generate the last image of the product set |
| Status | bool | Identify whether satellites and sensors are still working |
| Description | text | Introduction to the product set |

**Table 2.** 'Image Public Property' table structure.

| Field Name | Data Type | Field Description |
|:---:|:---:|:---:|
| Id | integer | Unique identification of the image |
| PId | integer | Unique identification of the product set to which the image belongs |
| GridCode | integer | Spatial index coding calculated according to image center coordinates and multi-layer grid levels |
| Cloud | double | Percentage of area covered by cloud |
| Atime | timestamp | Time the image was acquired |
| Band | integer | Band of the image |
| CenterPoint | point | Latitude and longitude of image center |
| UpperLeftPoint | point | Latitude and longitude of upper-left corner of image |
| UpperRightPoint | point | Latitude and longitude of upper-right corner of image |
| LowerLeftPoint | point | Latitude and longitude of lower-left corner of image |
| LowerRightPoint | point | Latitude and longitude of lower-right corner of image |
| FileSize | double | Size of image file |
| Downloads | integer | Number of image downloads |

The extraction of metadata mainly depends on file reading operations. Remote sensing image metadata from different sources may have different formats, such as Landsat metadata (TXT format) and MODIS metadata (XML format). Semantic heterogeneity exists between each source's remote sensing image metadata [31]. According to the metadata standard, we designed a set of algorithms for automatic metadata analysis and storage. Different file formats are analyzed by different methods. Locate the field position step by step and extract the field. The corresponding Hilbert code of the image is calculated according to the extracted spatial information. Finally, all metadata are output as a unified database record. The specific algorithm flow is shown in Figure 5.

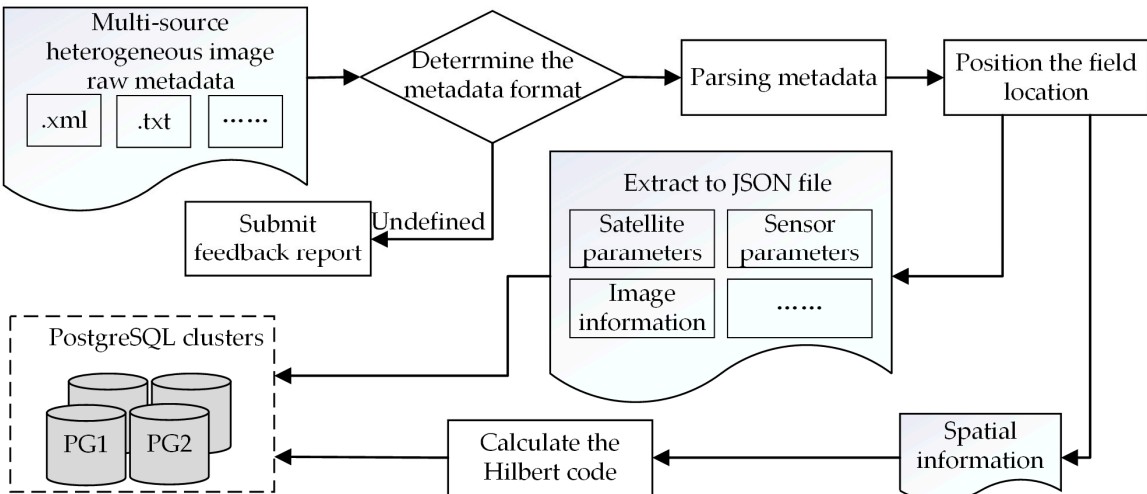

**Figure 5.** Algorithm flow of automatic extraction and storage of remote sensing image metadata.

### 2.2.3. Storage of Massive Image Tiles Based on an HDFS Distributed System

An HDFS is a distributed file storage system that runs securely and efficiently on a computer cluster using streaming data access mode to store massive datasets. Its most remarkable feature is that it can shield the computer hardware differences in the underlying cluster and present the cluster as a whole. A high-performance server is assigned as the Namenode of HDFS. The remaining computer nodes are Datanodes, and the number of Datanodes can be increased according to storage requirements to achieve capacity expansion. Namenode is used to store metadata of data blocks, and store and access data blocks through metadata. As the centralized metadata core of HDFS, Namenode manages and maintains metadata in memory to improve overall reading efficiency. Datanode is used to store the actual data. The default minimum storage space of HDFS is a block, and the default size of each block in Datanode is 128 MB.

When the user stores data, the Namenode splits the data according to the size of the pre-set block, and then stores the split data separately into Datanodes and makes a redundant backup. Then, Namenode will generate metadata to record the storage location of data blocks in Datanode and the mapping relationship between data blocks. When the user reads the data, the first request is made to the Namenode, which queries the mapping table to determine the location of the data block in the Datanodes. Then, all data blocks from the Datanodes are read out, and the data are re-assembled according to the mapping table [32,33]. Figure 6 shows a flowchart of HDFS data storage derived from Hadoop's official website [34].

The original file of the remote sensing image is stored in the HDFS file system. Due to the large amounts of remote sensing image data, if the HDFS file system only stores the original image, the hardware loading and network transmission performance cannot provide real-time image loading for browsing. Using the tile pyramid model can reduce the data rendered in the current view, thereby reducing image loading times and system memory loads. Pyramid construction and tile segmentation of remote sensing images are implemented by the Python script 'gdal_retile.py' in GDAL. This script can cut remote sensing images according to their basic information, such as the data source, starting coordinates in the x-y direction, generated pixel values, the data compression type used in slicing, the number of tile layers, and the generated target path.

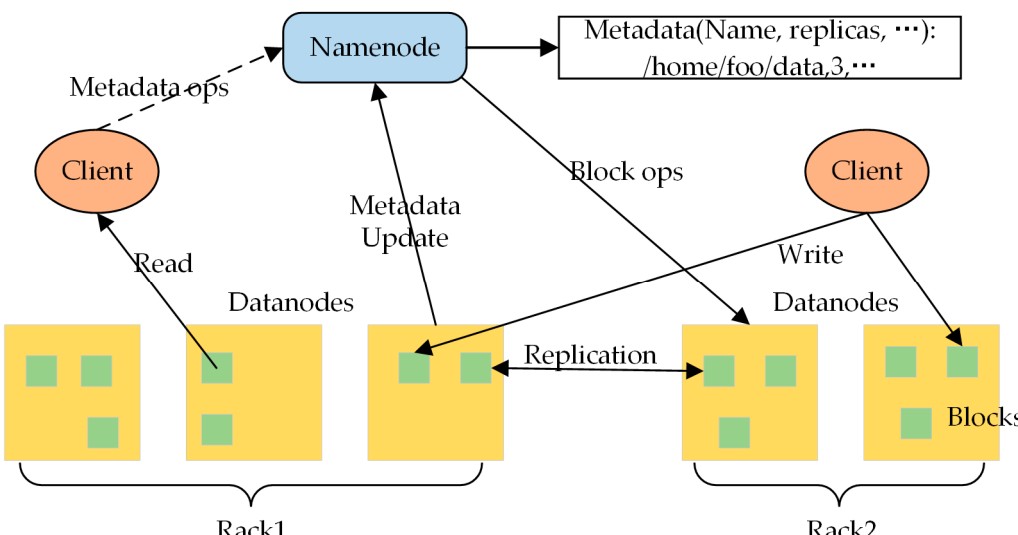

**Figure 6.** HDFS data storage process.

The lowest-resolution image tiles are stored at the top of the pyramid structure. In further pyramid layers, the resolution increases in turn. The original-resolution data are stored at the bottom of the pyramid. The whole pyramid structure is a series of images arranged in pyramids with resolutions from low to high. The hierarchical division of the image pyramid and the process of image size calculation are as follows. Assuming that the basic level of an image is J, and its size is $2^J \times 2^J$ or N × N (J = log$_2$N), then the size of the j-level in the middle of the pyramid is $2^j \times 2^j$, where $0 \leq j \leq J$. The complete pyramid consists of J + 1 resolution levels, with sizes ranging from $2^J \times 2^J$ to 1 × 1; that is, $2^0 \times 2^0$ (Figure 7). The total number of elements in the P + 1-level (P > 0) pyramid is:

$$N^2 \left( 1 + \frac{1}{4^1} + \frac{1}{4^2} + \cdots + \frac{1}{4^P} \right) \tag{1}$$

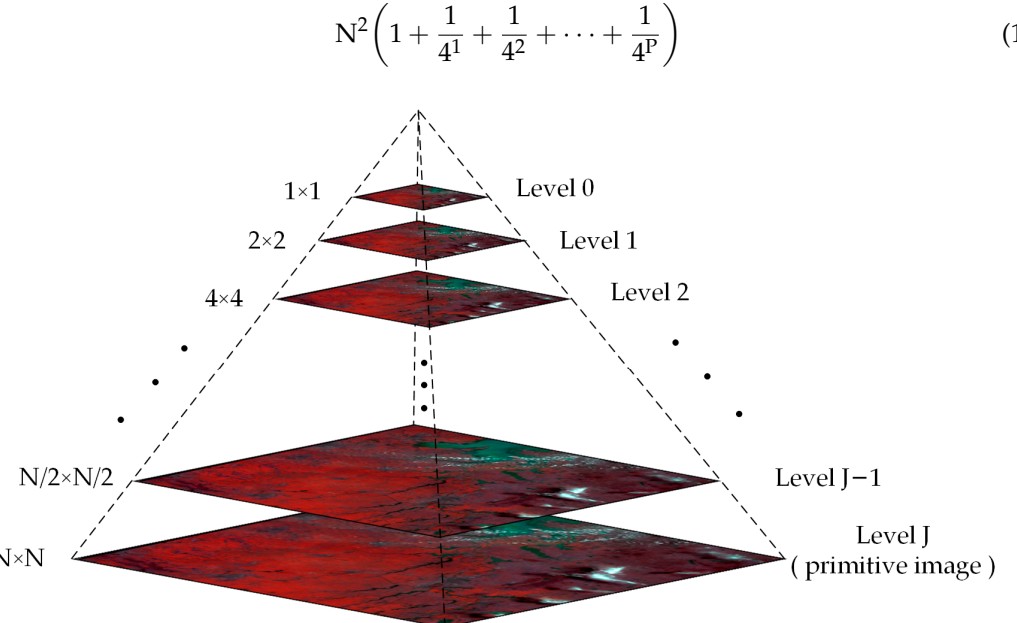

**Figure 7.** Structure of an image pyramid.

Hadoop Archive organizes the tile pyramid of remote sensing images. Hadoop Archive is a small file-merging method of Hadoop. The tiles of each image are packaged into a '.har' format file in Hadoop Archive and stored in HDFS. We create 'J + 1' subdirectories in the image storage file to store image data for each pyramid level and create another subdirectory under each directory to store tiles for each level of the pyramid image. The

tile data are segmented based on the row and column numbers, and the specific number of subdirectories depends on the number of tile rows in the pyramid image.

Establishing a pyramid and image tile segmentation can significantly improve the speed and performance of image preview but will take up more storage space. The pyramid is a collection of image versions with different resolutions. However, due to the substantial capacity scalability of the HDFS distributed file system, sacrificing storage space for better image preview is a worthwhile efficiency optimization strategy.

### 2.3. Scheduling of Remote Sensing Images

#### 2.3.1. Efficient Retrieval of Remote Sensing Images

Remote sensing image retrieval is based on multiple retrieval factors, including spatial location, time range, product set, cloud amount, etc. Spatial location retrieval depends on a spatial index. To further improve the performance of image retrieval, retrieval of the spatial location is also based on a multi-layer Hilbert grid, which transforms complex spatial intersection operations into simple coding matching (Figure 8). The specific steps are as follows:

1.  Wait for the user to submit the search conditions. The area of the retrieved region is denoted as S1. The Hilbert level corresponding to the product set is retrieved from the PostgreSQL cluster's 'Product Set' table, designated as L0. The grid area is denoted as S0;
2.  If S1 $\leq$ S0, the code of the L0-level Hilbert grid that intersects with the retrieval area is returned directly; if S1 > S0, we find the minimum Hilbert grid that can surround the search area and mark its grid level as L1.
3.  The L1-level grid is divided by a multi-layer Hilbert grid iteration, and the grid outside the retrieval area is eliminated in the division process. Once divided into L0-level grids, all L0-level Hilbert grid codes intersecting with the retrieval area are returned.
4.  The search factor submitted by the user and the calculated Hilbert code set is transmitted to the PostgreSQL cluster for query. Finally, the image sets retrieved from different PostgreSQL databases are collected, merged and returned to the client.

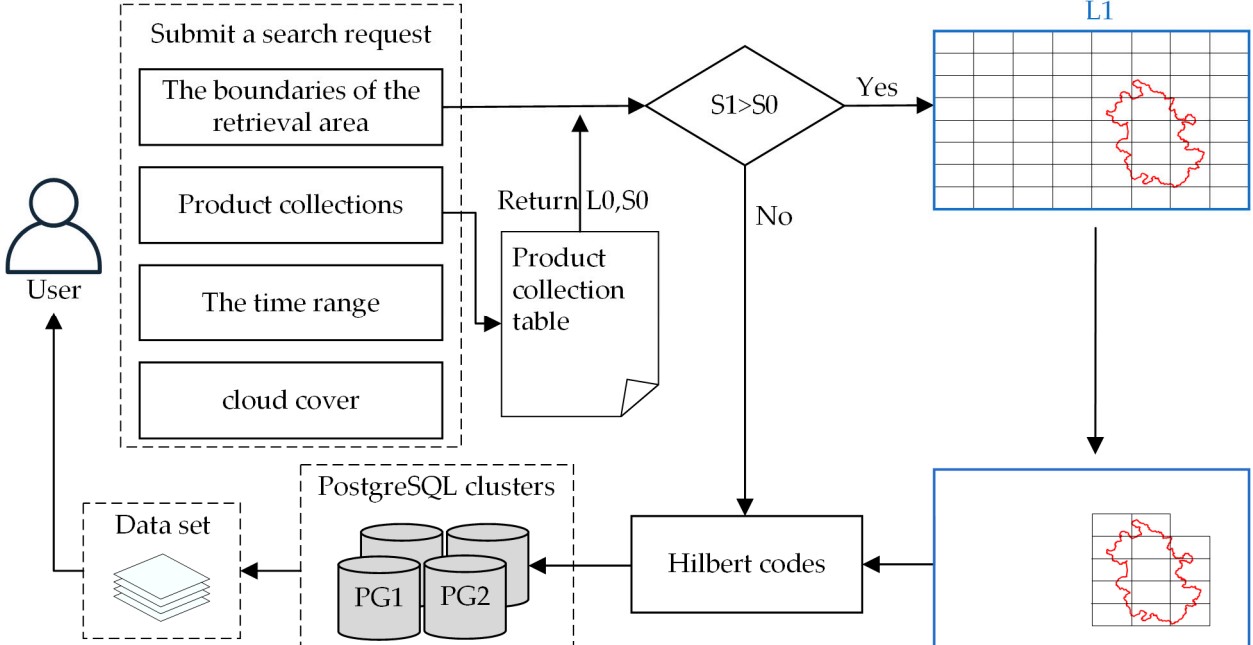

**Figure 8.** Image retrieval process.

### 2.3.2. Transmission of Remote Sensing Image Tiles

To load remote sensing image tiles, we first need to calculate the row and column numbers of the image tiles to be loaded according to the latitude and longitude range of the client's current view. Assuming that the current pyramid level is N, the degree of the partition interval between tiles at that level is D. The row and column number calculation process of the tiles to be loaded is as follows:

1. First, we obtain the zoom ratio of the current view and the screen coordinates of the bottom left and top right. Then, the screen coordinates are converted into geographic coordinates in the lower-left corner $(X_{LB}, Y_{LB})$ and upper-right corner $(X_{RT}, Y_{RT})$. The corresponding pyramid level N is calculated based on the view scaling.
2. Traverse the previously retrieved remote sensing images. If the image is outside the current view range, the tile of the image is not read. If the image intersects with the current view range, continue to the next step. If the image is entirely within the current view range, all tiles of the N-level pyramid of the image are read. In addition, to reduce unnecessary tile reading, only the latest image tiles in the retrieval time range are read for the image overlap area in the current view.
3. Calculate the row and column numbers of the N-level pyramid tiles in the lower-left corner of the current view based on $(X_{LB}, Y_{LB})$:

$$\begin{cases} col_{LB} = \frac{X_{LB}+180^{\circ}}{D} \\ row_{LB} = \frac{Y_{LB}+180^{\circ}}{D} \end{cases} \tag{2}$$

4. Calculate the row and column numbers of the N-level pyramid tiles in the upper-right corner of the current view according to $(X_{RT}, Y_{RT})$:

$$\begin{cases} col_{RT} = \frac{X_{RT}+180^{\circ}}{D} \\ row_{RT} = \frac{Y_{RT}+180^{\circ}}{D} \end{cases} \tag{3}$$

5. Read the image tiles in the collection $\{(col_{LB}, row_{LB}),(col_{RT}, row_{RT})\}$ from the HDFS distributed file system.

### 2.3.3. Visualization of Remote Sensing Images

Visualization of remote sensing image tiles is achieved through the Leaflet library (Figure 9), which is an open-source online mapping JavaScript library developed by Vladimir Agafonkin's team. Although its source code is only 33 KB, it has most of the functionality required for developing online maps [35,36]. The user view mainly includes map translation, scaling, browsing, image tile display and image metadata query functions.

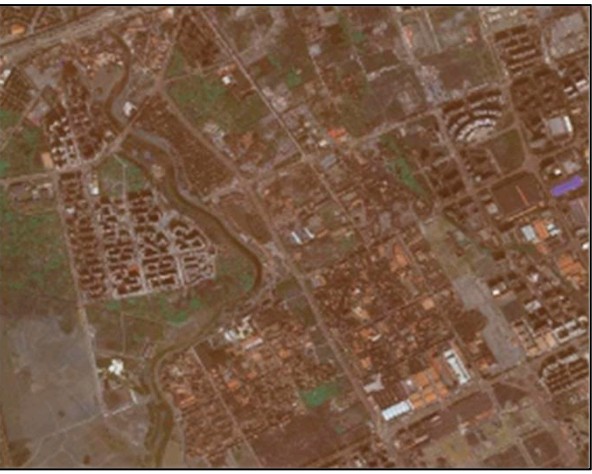

**Figure 9.** Remote sensing image tiles loading using Leaflet.

### 2.4. Strategies for Optimizing Image Scheduling Efficiency

Section 2.3 describes the core method of remote sensing image scheduling. The scheduling process is divided into three steps:

1. According to the data request, retrieve the required image dataset from the PostgreSQL database cluster;
2. Retrieve image tiles from the HDFS file system through the I/O interface;
3. Visualize and render the image tiles through Leaflet and present the final view to the client.

One can achieve faster image data scheduling through these steps, but it is still not a real-time second-scale response. To further optimize the scheduling efficiency, this section describes measures that improve data retrieval, transmission, visualization and other aspects, including mechanisms for image tile prefetching, multi-threading and ring caching.

#### 2.4.1. Ring Caching Mechanism

When scheduling images, the computer will continue to allocate and release memory when performing many read and write operations. The memory size of each allocation and release is inconsistent. Frequent memory operations can lead to a large amount of fragmented memory, resulting in a gradual decrease in CPU access to memory efficiency, and computer stalling [37]. Therefore, we designed a multi-caching ring buffer design to display images. The core principle is to apply for continuous memory so that data storage and release occur within a closed-loop memory address. This design can handle a steady stream of data without continuing to apply for new memory space to temporarily store new data (Figure 10). After the remote sensing images are segmented and resampled, the data size of each image tile is about 1~2 MB. The primary purpose of the ring caches is to act as a data transfer station for loading image tiles. Therefore, we set the size of each memory block to 3 MB to ensure that the image tiles are scheduled correctly.

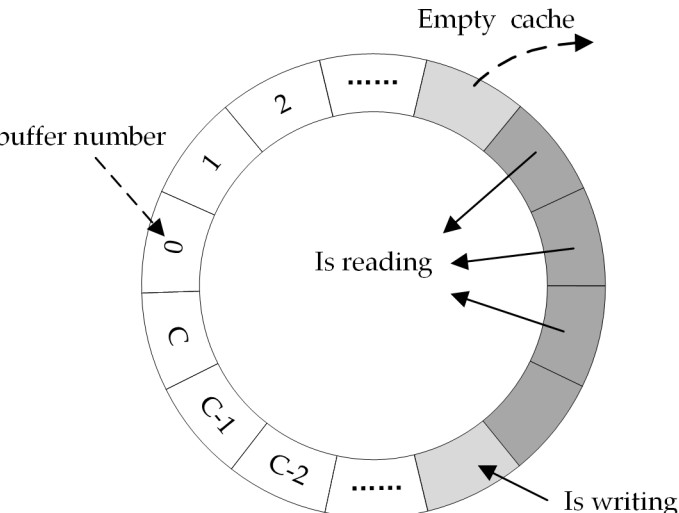

**Figure 10.** Ring caching mechanism.

To ensure the correctness of the input/output order in the ring buffer, two types of pointers are set for the ring buffer—a *read* pointer and a *write* pointer. The *read* pointer points to the data being read in the ring buffer, while the *write* pointer points to its writable memory. Moving the *read* and *write* pointers ensures that the order of reading and writing in the buffer is not disordered. The pointer movement process of the ring caching mechanism depends on the multi-threading mechanism, and their specific interaction methods will be introduced in Section 2.4.3. A traditional chain buffer may allocate new storage space to store new data when writing. During readout, the storage space of discarded elements may be released. All storage space in the ring buffer is allocated in advance. All write and

read operations are carried out within a fixed storage space. The data are emptied for the cache where the data have been read, but the storage space is not released. So, compared to an ordinary chain cache, the ring cache avoids frequent allocation and release of storage space during the data transmission process, improving the data scheduling efficiency and storage stability.

### 2.4.2. Tile Prefetching Mechanism

Users may frequently perform translation and scaling operations when previewing images. The view needs to perform dynamic data scheduling according to user viewpoint changes. However, suppose the data scheduling process is restarted when the required data have entered the view range. In that case, the preview process will inevitably stagnate, making it fail to meet the real-time requirements. Therefore, the image tile prefetching problem must be considered.

Inspired by the binary tree traversal algorithm, the image tile prefetching mechanism is divided into breadth-based prefetching and depth-based prefetching.

- Breadth-based prefetching is used to prefetch image tile data in the scene of the view translation operation. The purpose is to read the image tile data spatially adjacent to the current view area and transfer it to the cache area in advance. Figure 11 shows a schematic diagram based on the breadth prefetching model. The grey area represents the current view area range, while the white area represents the prefetching range for the tiles. The side length of the prefetching range is twice the side length of the current view range.

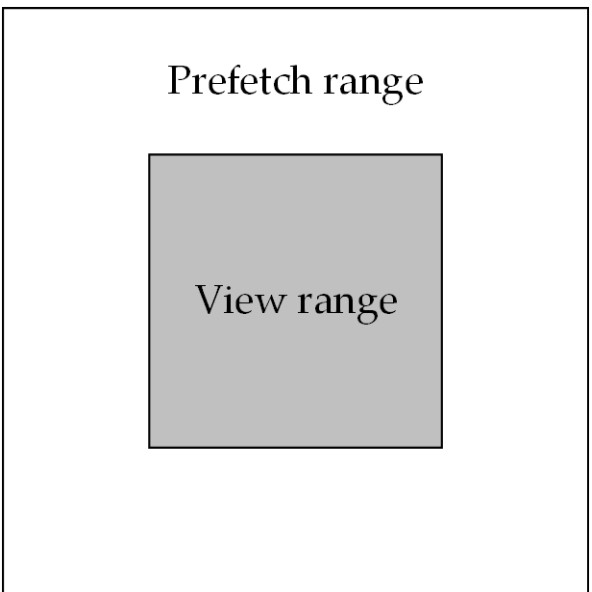

**Figure 11.** Schematic diagram based on the breadth-based prefetching model.

- Depth-based prefetching is used for image tile data prefetching under map zoom operations. The purpose here is to read the image tile data of the current view area from the adjacent zoom levels and transfer it to the cache in advance. A schematic diagram based on the depth-based prefetching model is shown in Figure 12. The left side of the image is the image pyramid. Assuming that the image pyramid level corresponding to the scaling ratio of the current view is k, the k + 1-level and k − 1-level image pyramid tiles are prefetched from the storage system to the cache.

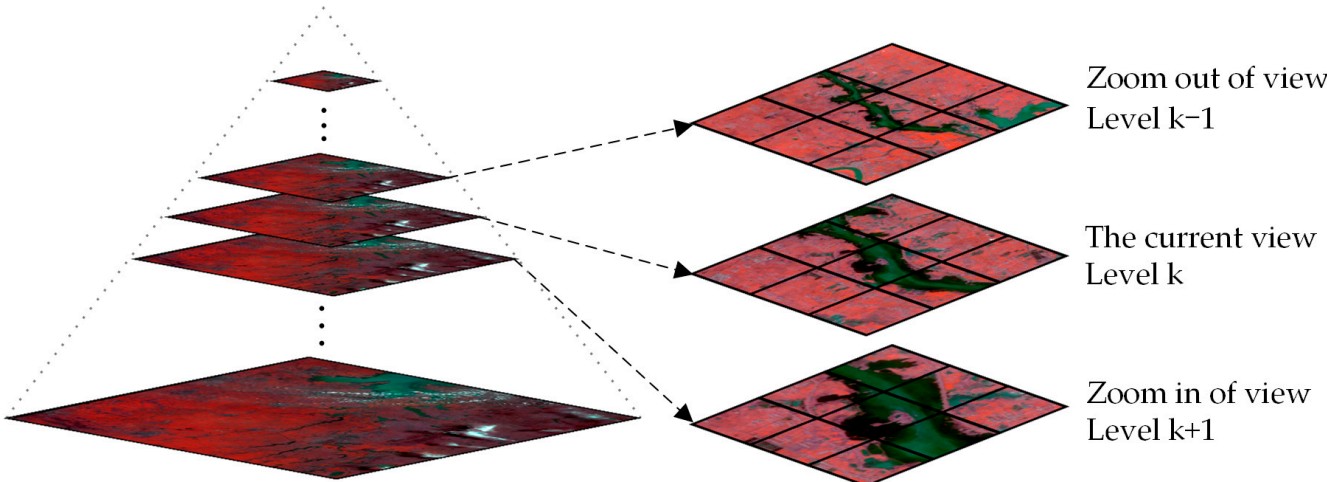

**Figure 12.** Schematic diagram based on the depth-based prefetching model.

To respond quickly when the user sends a signal that the view range changes, we divide the roaming direction of the view into ten categories. These comprise the eight directions of view plane roaming, view reduction and view amplification. The prefetched tile blocks are stored in ring memory according to these ten categories. The prefetched tiles in each category are stored in adjacent memory locations, and pointers are created at the starting position of each block. When the view range changes in a particular direction, the cache position of the image tile in that direction is quickly located by a pointer.

### 2.4.3. Multi-Threading Mechanism

The scheduling process is further optimized using a multi-threading mechanism to maintain image scheduling efficiency and achieve smooth image data visualization. Multi-threading refers to a computer using multiple threads to perform concurrent operations based on software or hardware, thereby accelerating processor efficiency [38]. The image scheduling process is divided into three threads: an image retrieval thread, an I/O thread and an image visualization thread.

The image retrieval thread is responsible for retrieving the attribute information of a desired image from the PostgreSQL database cluster based on the front-end data request. As shown in Figure 13, to improve the concurrency and capacity scalability of the PostgreSQL database cluster, differently encoded image metadata are stored in different databases according to the 3-level Hilbert grid and the 64 databases are deployed to four servers according to the 1-level Hilbert grid division. Each server in the cluster is accessed through a sub-thread. When the data requested by the user are stored in a server, the sub-thread corresponding to the server is awakened to perform data retrieval according to the spatial index.

The I/O thread is responsible for reading the image tile data required by the current view, which needs to be prefetched into the ring buffer from the HDFS distributed cluster. When the I/O thread reads the data, it first needs to obtain the *DistributeFileSystem ()* instance, which calls an RPC mechanism to access the Namenode through the *open ()* method to obtain the description and location information of the requested data block. Then, the information obtained is read to the Datanodes.

**Figure 13.** Deployment method for the PostgreSQL database cluster.

The image visualization thread is responsible for reading the image tile data from the ring buffer to the user view. When the user retrieves image data, the image tile data are read into the first memory in the ring buffer by the I/O thread to wake up the image visualization thread. Then, the pointer to the cache address is passed to the image visualization thread. The image visualization thread then begins reading the data to the user view through the pointer. When the user view range changes, the image visualization thread returns the pointer of the data removed from the view range to the I/O thread, and the I/O thread clears the cache of these data. At the same time, the image visualization thread finds the available prefetched data in the ring buffer and reads it to the view. Finally, the image visualization thread submits the read data information to the I/O thread, so that the I/O thread reduces the read amount of the new round of prefetched data.

As the image retrieval thread, I/O thread and image visualization thread are not executed at the same time, the I/O thread should wait for the memory storage space to be empty before writing data, while the image visualization thread needs to wait for the space to not be empty before reading data. A pointer tunes these two waiting processes. Two types of pointers for the ring buffer are set: *'Is writing'* and *'Is reading'*.

*'Is writing'*: If the identifier is 'True', the cache is being written to data by the I/O thread. Therefore, if the image visualization thread reads the data too fast, it will be paused at the cache. After waiting for the I/O thread to write and reset the 'Is writing' identifier to 'False', the image visualization thread can read the data in the cache area.

*'Is reading'*: If the identifier is 'True', the data in the cache are being used by the image visualization thread. At this time, the I/O thread cannot read and write to the memory buffer. When the data are moved out of the view range, and the I/O thread receives the cache emptying instruction, the I/O thread resets the identifier to 'False' and empties the cache. The I/O thread can then write data to the cache.

Figure 14 shows the interaction between the multi-threading and ring caching mechanisms. The *'Is writing'* and *'Is reading'* pointers ensure that the multi-threading mechanism does not mess up the process of dispatching data from the ring caches. This does not mean that the two threads are sequential, but are interleaved. When the I/O thread reads the data and stores it in the hollow memory space of the ring cache, the memory is passed to the image visualization thread. The I/O thread itself continues to read data from the HDFS distributed system and then stores it in other empty memory spaces in the ring buffer. In such a scheduling process, the I/O thread always writes and leaves to the empty memory of the ring cache, and the image visualization thread reads and leaves from the non-empty memory. Thus, fast data scheduling is achieved.

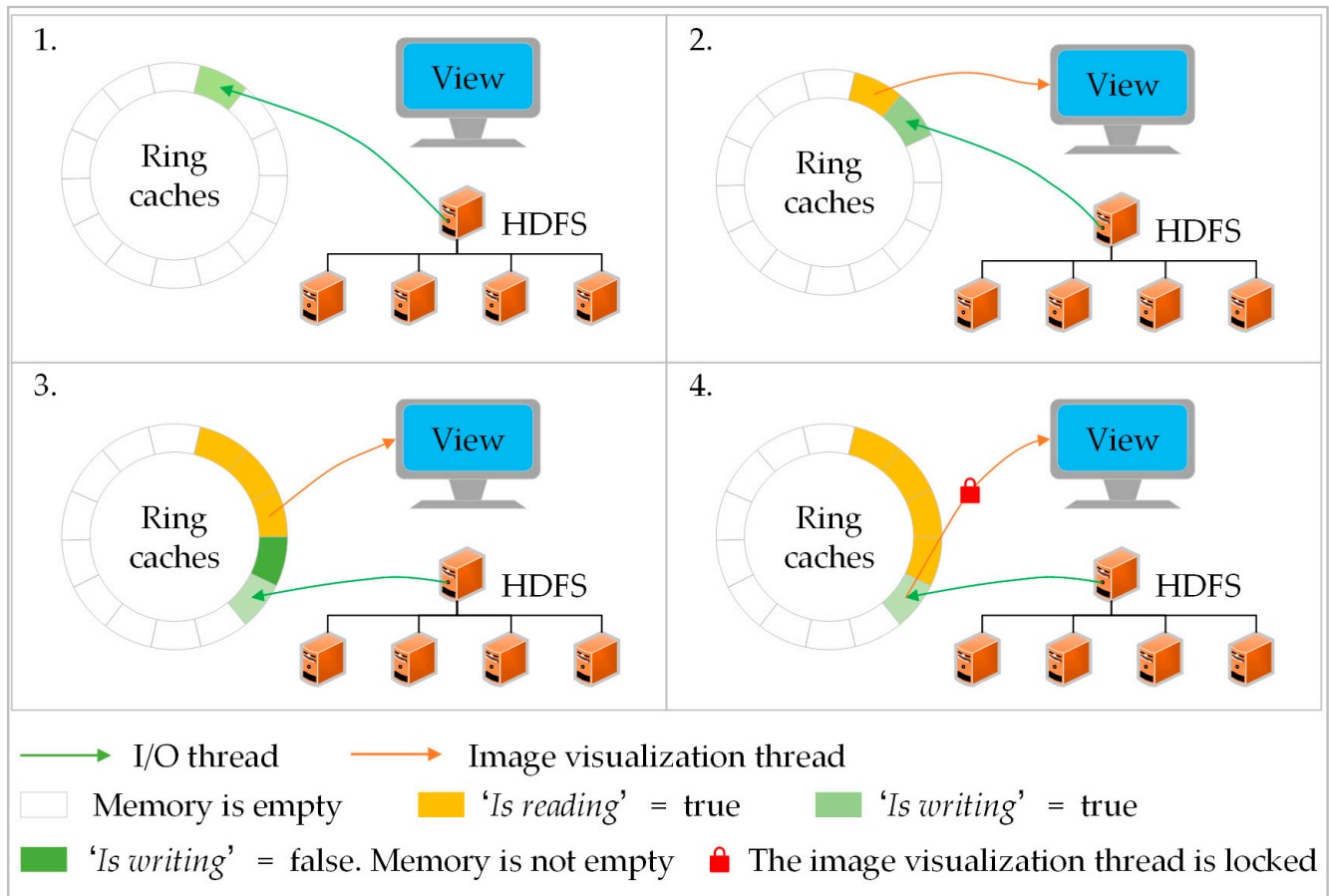

**Figure 14.** The diagram of the interaction between the multi-threading mechanism and the ring cache mechanism(Figure 1 shows the beginning of writing data, Figure 2 shows the beginning of reading data, Figure 3 shows the process of reading and writing data, Figure 4 shows that the data is read out too fast, and the image visualization thread is locked).

## 3. Results

### 3.1. System Testing

#### 3.1.1. Test Environment Deployment

In this experiment, 14 virtual machines were created by VMware software to build a system cluster. The configuration of the virtual machines is shown in Table 3. Among them, one was used to build the system server, four were used to create the PostgreSQL database cluster, one was used to build the Namenode of the Hadoop cluster, and eight were used to build the Datanodes of the Hadoop cluster. All virtual machines were on the same LAN. The environment was limited to system testing. A complete set of fully distributed Hadoop clusters often requires several to dozens of dedicated high-performance servers as computer nodes and large-scale high-speed networks must be deployed to achieve interconnection of various nodes.

**Table 3.** Virtual machine configuration parameters.

| Purpose of the Virtual Machine | Quantity | Parameters for the Virtual Machine |
|---|---|---|
| Web Server | 1 | CPU: 11th Gen Intel$^{(R)}$ Core$^{(TM)}$ i7-11700 @ 2.50 GHz, Memory: 16 GB, HDD: 200 GB, NIC: 1000 Mbit/s, OS: Windows 10 |
| PostgreSQL Clustering | 4 | CPU: 11th Gen Intel$^{(R)}$ Core$^{(TM)}$ i7-11700 @ 2.50 GHz, Memory: 4 GB, HDD: 200 GB, NIC: 1000 Mbit/s, OS: CentOS 7 |
| Namenode | 1 | CPU: 11th Gen Intel$^{(R)}$ Core$^{(TM)}$ i7-11700 @ 2.50 GHz, Memory: 4 GB, HDD: 200 GB, NIC: 1000 Mbit/s, OS: CentOS 7 |
| Datanodes | 8 | CPU: 11th Gen Intel$^{(R)}$ Core$^{(TM)}$ i7-11700 @ 2.50 GHz, Memory: 2 GB, HDD: 200 GB, NIC: 1000 Mbit/s, OS: CentOS 7 |

### 3.1.2. Experimental Data

Since collecting and downloading remote sensing images requires much time and storage space, this experiment only collected remote sensing images from three datasets: Landsat 5 TM, Landsat 8 OLI/TIRS and Sentinel-2A MSI. A total of 252 GB of images was obtained to test the storage and image scheduling performances of RSIMSS. To test the retrieval performance of RSIMSS with massive image sets, the experiment virtualized 10,000,000 image metadata in batches according to the metadata structure of the three image datasets and stored them in the PostgreSQL cluster. The virtual image metadata were evenly distributed worldwide. The information on the experimental data are shown in Table 4.

**Table 4.** Experimental data.

| Datasets | Resolution (m) | Time Coverage | Dimensions (km$^2$) | Hilbert Grid Level |
|---|---|---|---|---|
| Landsat 5 TM | 30 | 1 January 1990–31 December 2010 | 185 × 185 | 6 |
| Landsat 8 OLI/TIRS | 30 | 1 January 2014–31 December 2020 | 185 × 185 | 6 |
| Sentinel-2A MSI | 10, 20 | 1 January 2018–30 June 2022 | 290 × 290 | 6 |

### 3.1.3. Storage Performance Testing of the RSIMSS

To test the storage performance of RSIMSS, a CEPH distributed storage system was selected as a reference for comparison, which is a popular distributed storage system. It uses the CRUSH algorithm to realize decentralized distributed storage. It is suitable for storing massive datasets and has uniform data storage distribution and high parallelism [39].

With RSIMSS and CEPH deployed under the same hardware conditions, the following two tests were constructed to evaluate the performance of their image data storage.

Test 1: Under the same client-server, 5, 10, 20, 50, 75, or 100 images were randomly selected. According to the same order, RSIMSS and CEPH were used to upload and download images, respectively, and the average upload and download speeds were recorded. The test results are shown in Figure 15.

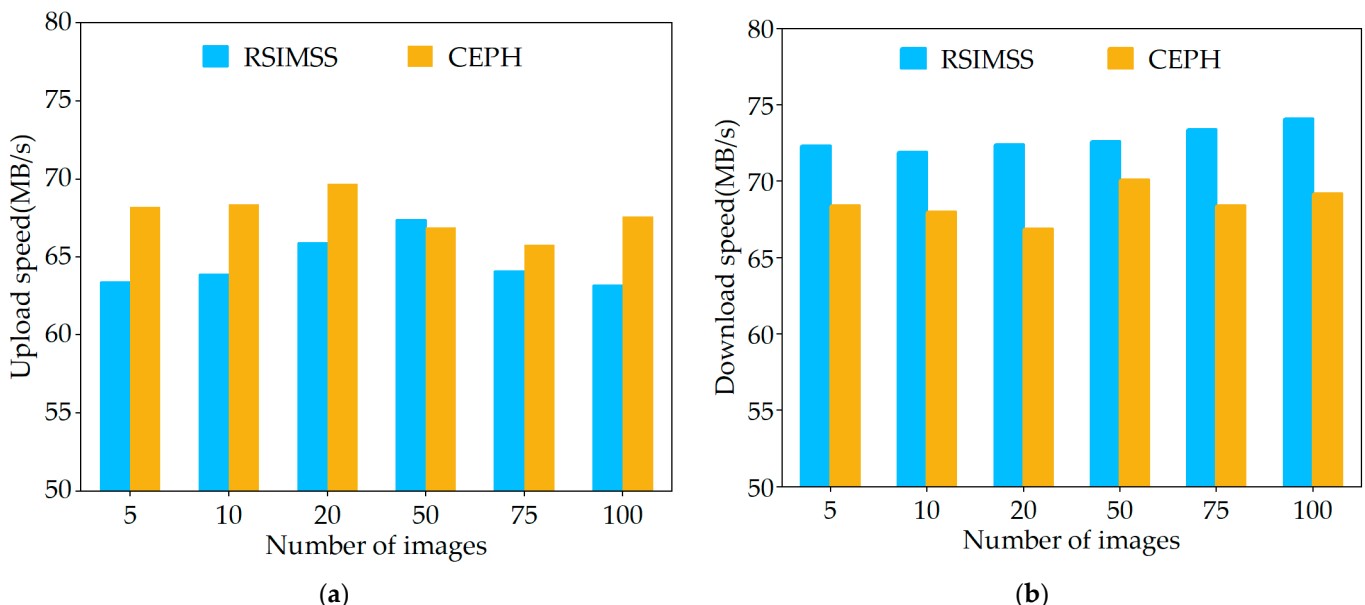

**Figure 15.** Results of Test 1: Image upload (**a**) and download (**b**) speeds by the RSIMSS and CEPH systems.

Test 2: This was used to test the concurrency of RSIMSS and CEPH by simulating multiple users through concurrent I/O processes. Their input and output performances were tested under 10, 20, 50 and 100 concurrent I/O processes. The image data read by each thread were 1 GB in size. The test results are shown in Figure 16.

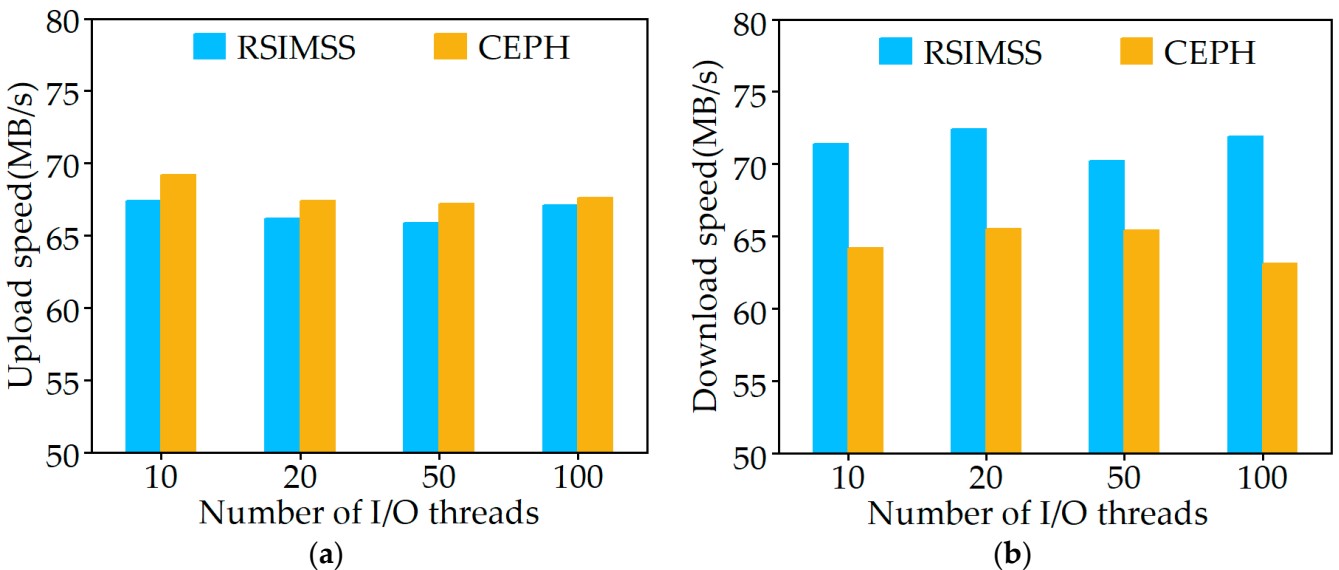

**Figure 16.** Results of Test 2: Image upload (**a**) and download (**b**) speeds by the HDFS and CEPH systems during multi-process access.

The test results show that RSIMSS had slightly lower data upload speeds than CEPH, but the former data download speed was better than the latter. In the case of multi-process access, with increases in the number of I/O processes, the image upload/download speeds of RSIMSS and CEPH remained stable, which proves that both have excellent concurrency. However, under the storage mode with remote sensing image single entry and multi-user concurrent access, the download speed requirement is much higher than the upload speed. Therefore, RSIMSS is more suitable for remote sensing image storage than CEPH. Due to

the experimental environment's limitations, testing the capacity scalability and stability of the storage system was challenging. However, Arafa [40], Peter [41] and others have done research in this area. Although the storage capacity of the CEPH system can be expanded infinitely, this process is not smooth. The change in Crushmap during each expansion will lead to CEPH rebalancing, with frequent changes impacting its internal stability. The underlying distributed file system HDFS in RSIMSS adopts a centralized metadata structure. Datanodes are almost unlimited in capacity expansion. Blocks in Datanodes will back up copies at other nodes. Even if a failure occurs, it does not affect the external service of the cluster, so the system stability is high.

3.1.4. Image Retrieval Performance Test of the RSIMSS

This experiment tested how RSIMSS improved the spatial retrieval performance of remote sensing images through the multi-layer Hilbert spatial index algorithm. The retrieval efficiencies of the multi-layer Hilbert spatial index and BRIN spatial index algorithm in PostGIS were compared under the same server environment. PostGIS is an extension module of the PostgreSQL database, which follows the Open GIS specification and provides functions such as spatial object building, spatial data indexing, spatial data manipulation functions, and spatial data operators. The two most widely used spatial index algorithms in PostGIS are GIST and BRIN. The BRIN index is chosen as the contrast reference because the GIST index has a complex structure and large storage space, making it more suitable for spatial operations with complex spatial relations. BRIN is more suitable for spatial data retrieval when there are large amounts of data and continuous spatial distribution and is more suitable for retrieving remote sensing images.

To compare the image retrieval performance of the multi-layer Hilbert spatial index and BRIN spatial index in PostGIS, three polygonal regions were selected for image retrieval (Figure 17).

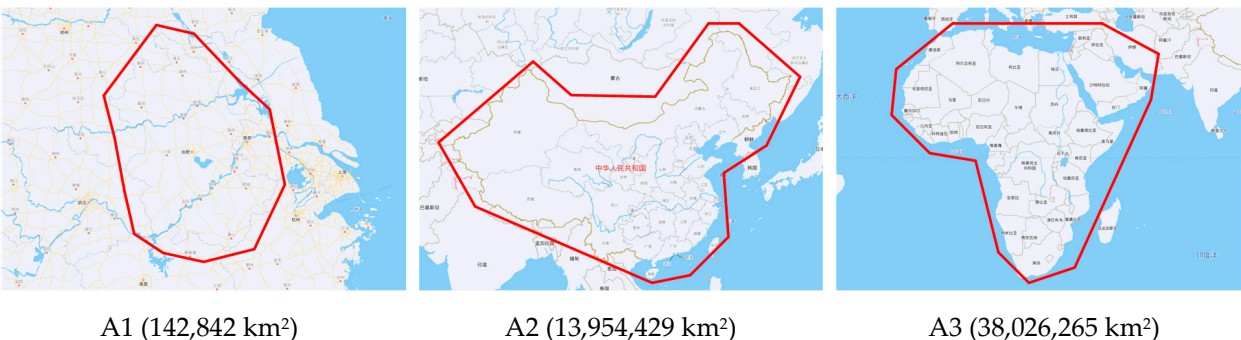

A1 (142,842 km²)  A2 (13,954,429 km²)  A3 (38,026,265 km²)

**Figure 17.** Areas used for the retrieval performance test.

The three retrieval regions are from different global locations and have different sizes. Region A1 is the smallest and only lies in a single Level 3 Hilbert grid, where the imagery metadata are stored in the same database. Region A2 spans multiple Level 3 Hilbert grids but is only in a single Level 1 Hilbert grid, and the imagery metadata for the area are stored in multiple databases on the same server. Region A3 is the largest and spans multiple Level 1 Hilbert grids, with the imagery metadata stored on different servers in the database cluster.

Test 3: To test the image retrieval performance, a multi-layer Hilbert spatial index and BRIN spatial index were used to retrieve images from the above three regions. The time consumed is shown in Figure 18.

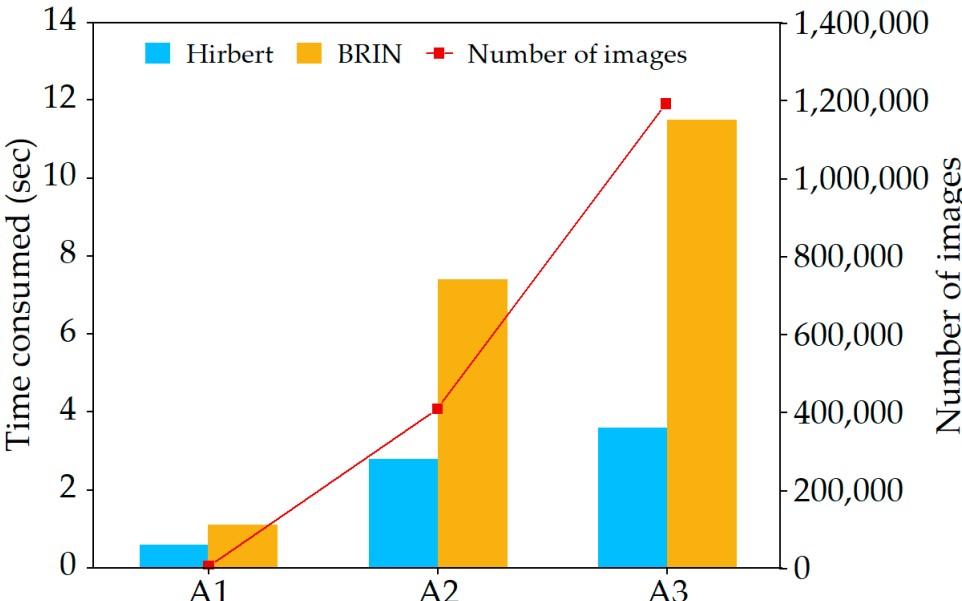

**Figure 18.** Results of Test 3: Time consumed to retrieve images from Regions A1, A2 and A3, and the number of images retrieved.

The Test 3 data show that the number of images retrieved from Region A1 was 6607, and the Hilbert index and BRIN index retrieval times were 0.6 s and 1.1 s, respectively. The retrieved data increased from 6607 in A1 to 1,193,044 in A3, the Hilbert index retrieval time increased to 3.6 s and the BRIN index retrieval time increased to 11.5 s. The test results show that the retrieval efficiency of the multi-layer Hilbert spatial index designed in this paper is better than the BRIN index. The advantage becomes more evident with increasing amounts of retrieved data.

### 3.1.5. Image Scheduling Performance Test of the RSIMSS

The RSIMSS adopts multi-threading, ring cache and prefetching mechanisms to optimize image scheduling. These three optimization mechanisms must be started sequentially, as subsequent mechanisms can only be started if the previous mechanism has started. To test the improvement conferred by each mechanism to image scheduling performance, we componentized them in the service layer of the RSIMSS and controlled their activity via interfaces. The invocation status of the components can be divided into three cases:

- Case 1: All three optimization mechanisms are closed;
- Case 2: Only the ring cache mechanism is enabled;
- Case 3: Simultaneously start the ring cache mechanism and the multi-threading mechanism;
- Case 4: All three optimization mechanisms are started.

The following two tests were designed in this experiment.

Test 4: Cases 1, 2 and 3 were used to test the image scheduling efficiency of RSIMSS: a total of 20 image tiles were scheduled in different user views. After the image retrieval was completed, the timer was set at the image tile read from the HDFS cluster. Statistical image tiles take time from the start of reading to full display in the user view. The test results are shown in Figure 19.

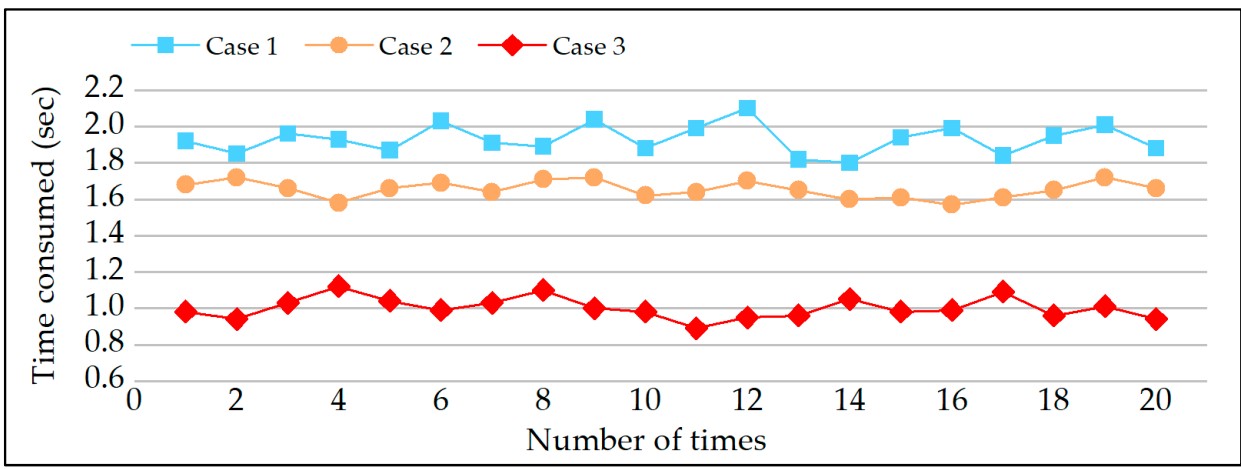

**Figure 19.** Results of Test 4: Comparison of image scheduling efficiency of RSIMSS in cases 1, 2 and 3.

Test 5: The view roaming efficiency of RSIMSS is tested by cases 3 and 4. This test performed pan, zoom-in, and zoom-out operations 20 times. The elapsed time between mouse release and refreshing the user's view was recorded. The results are shown in Figure 20.

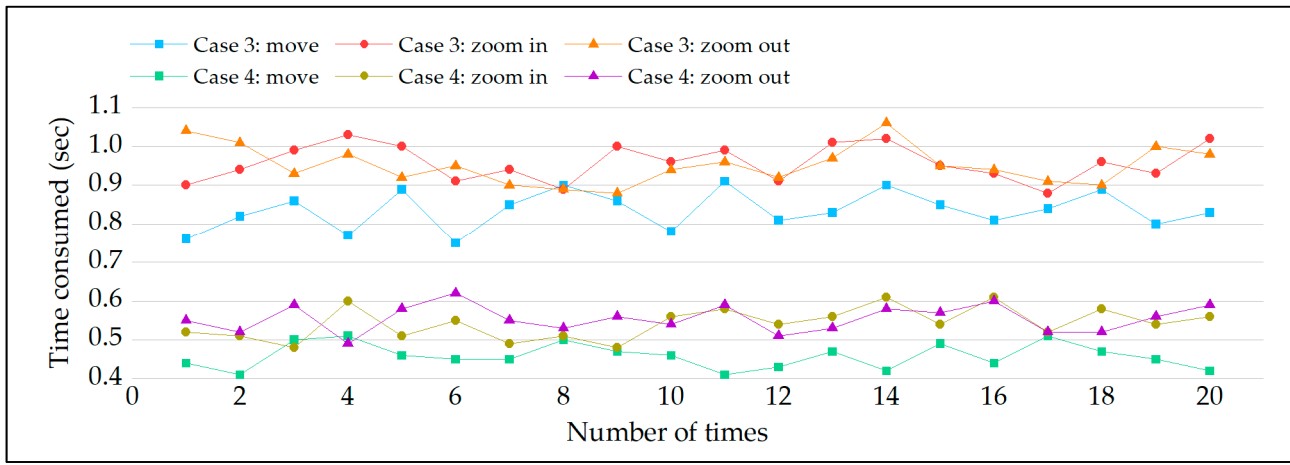

**Figure 20.** Results of Test 5: Comparison of view roaming efficiency of RSIMSS in cases 3 and 4.

The results of Test 4 show that the average times for image tile scheduling in cases 1, 2 and 3 were 1.93 s, 1.66 s and 1.00 s, respectively. The scheduling time was reduced by 0.27 s when only the ring cache mechanism was started and by 0.93 s when both the ring cache and multi-threading mechanisms were started. The results of Test 5 show that the average times taken for panning, zooming in and zooming out of the view in case 3 were 0.84 s, 0.96 s, and 0.95 s, respectively, while the average times in case 4 were 0.46 s, 0.54 s, and 0.56 s. With the support of a 1000 Mbit/s network card, the prefetching mechanism reduces the view roaming time to about half a second, realizing real-time roaming response.

### 3.2. Discussion of Test Results

Section 3.1 describes RSIMSS performance tests using remote sensing imagery from different data sources, including 252 GB of image files and 10,000,000 pieces of image metadata. A total of five experimental tests were carried out, with the results showing that RSIMSS provides efficient and stable image storage, retrieval and scheduling performance.

With increases in the number of Earth observation satellites, the number of remote sensing images has gradually increased. Theoretically, the capacity of the remote sensing

image storage system must be indefinitely expandable. The remote sensing image files and tile pyramid of RSIMSS are stored in the HDFS distributed cluster. HDFS is based on using the Namenode of centralized metadata to control the storage system as a whole. The Datanodes can be extended to any number of locations, and communication and data transmission between nodes can be undertaken through the network. Many nodes form a file system network. This architecture makes HDFS almost unlimited in capacity scalability, making it suitable for mass remote sensing image data storage. In use, users do not need to care about the image storage details and only need to manage and store image data through the Namenode. Tests 1 and 2 compared the image input/output performance and concurrency of RSIMSS and CEPH. The test results show that although RSIMSS and CEPH have higher storage performance, HDFS has higher system stability, making it more suitable for storing massive image datasets with a single entry and multi-user concurrent access.

All the metadata of the remote sensing images are stored in the PostgreSQL database cluster. The automatic metadata extraction and storage algorithm is customized according to each data source's metadata characteristics, dramatically reducing the workload and avoiding the human error risk of manual storage. We designed a multi-layer Hilbert spatial index algorithm based on a Hilbert space-filling curve. According to the characteristics of Hilbert grid segmentation, the global image metadata are evenly stored in 64 databases on four servers in the PostgreSQL cluster, which improves the cluster's capacity scalability and I/O concurrency. The retrieval algorithm converts complex spatial intersection operations into simple code matching in image retrieval. The multi-level Hilbert grid division method enables the system to quickly filter out a large number of images when performing spatial queries and achieve accurate retrieval area positioning. Thanks to the extremely high locality retention of the Hilbert curve, deploying the PostgreSQL cluster according to the Hilbert grid division method avoids cross-server data retrieval as much as possible. Even when this cannot be avoided, retrieval efficiency can be improved by using multi-threaded parallel retrieval. According to the Test 3 outcomes, the RSIMSS can retrieve millions of images in seconds. The multi-layer Hilbert spatial index has higher and more stable spatial retrieval performance than the BRIN index, making it more suitable for retrieving massive remote sensing image datasets.

The image scheduling process is divided into three parts: image tile retrieval, transmission and visualization. In Section 2.4, ring cache, multi-threading and tile prefetching mechanisms were designed to optimize the scheduling process. Although the optimization strategy is divided into three parts, these three are closely linked, and it is not a simple sequential execution process. As the basis of the entire image scheduling process, the ring caching mechanism is the data transfer station for image transmission, image reading, and tile prefetching. The ring cache is a contiguous memory allocated in advance, eliminating the frequent allocation and release of storage space during data reading, avoiding memory fragmentation, and improving system stability. The multi-threading mechanism acts as a regulator of the entire image-scheduling process. It divides the writing and reading of image tiles in the ring cache into two parallel processes, accelerating the scheduling efficiency. Two pointers control the correct order of image tile writing and reading to ensure that the scheduling process is not disordered. The image tile prefetching mechanism optimizes the view roaming efficiency of the image tiles. It reads the image tiles adjacent in space and resolution to the current view and stores them in the ring cache in advance. It then quickly locates the storage location of the prefetching tiles through the pointer when the user makes roaming instructions. According to the Test 4 and 5 data, the three optimization mechanisms work together to greatly improve the image tile scheduling efficiency, allowing real-time data scheduling and view roaming.

In summary, RSIMSS has excellent storage, management, retrieval and scheduling performance of massive remote sensing images. Its advancement and advantages mainly come from the following aspects: (1) According to the data format characteristics of remote sensing images, a HDFS distributed file system and PostgreSQL relational database are used to hybrid manage remote sensing images, so that RSIMSS has both the powerful

transaction processing ability of relational databases and the massive data throughput ability of distributed file systems. (2) Aiming to evaluate the spatial characteristics of remote sensing images, a multi-level Hilbert grid coding is designed and applied to spatial index construction, rapid retrieval area location and PostgreSQL cluster deployment. RSIMSS converts complex spatial intersection operations into simple coding matching, substantially improving the spatial retrieval performance of multi-source heterogeneous remote sensing images. Notably, multithreading mechanisms realize fast cross-database and cross-server queries. (3) RSIMSS is not limited to the traditional spatial index algorithm and tile pyramid technology to optimize the efficiency of image scheduling. At the same time, according to the data transmission characteristics of remote sensing images in the computer network, three new optimization mechanisms (ring cache, multithreading and tile prefetching mechanism) are used to achieve real-time scheduling and data roaming.

## 4. Conclusions

Remote sensing images are increasingly important in environmental protection, disaster prevention and engineering. They have become one of the most critical data sources in many industries. However, managing and scheduling large amounts of remote sensing imagery remains a considerable challenge. This study designs and develops a new remote sensing image management system (RSIMSS) to achieve efficient storage and real-time scheduling of massive remote sensing image datasets.

Distributed file systems have gradually emerged due to their powerful capacity expansion capabilities and data throughput, becoming one of the mainstream methods to store massive remote sensing images. However, the data retrieval method of the distributed file system is based on the file, and it is difficult to identify the specific attribute information inside the file. Therefore, realizing the attribute retrieval and spatial retrieval of multi-source heterogeneous remote sensing images is difficult. RSIMSS uses the relational database and the distributed file system to manage remote sensing images in a hybrid manner. The PostgreSQL database is used to manage the structured metadata of remote sensing images. A unified data organization integrates and stores the multi-source heterogeneous image metadata. The spatial index is established according to the multi-layer Hilbert grid to improve the spatial retrieval performance of the image. HDFS stores unstructured file data of remote sensing images with high I/O performance and almost unlimited capacity scalability. RSIMSS gives full play to the advantages of relational databases and distributed file systems, providing a new mode for storing and managing massive remote sensing image data.

Scheduling remote sensing images is more computer-oriented research, and its process relies on computer hardware and network transmission. Presently, most of the research on remote sensing image scheduling efficiency optimization methods is regarding the spatial index algorithm and the tile pyramid model, which have not been further explored. This paper analyzes the data transmission process of remote sensing images between the storage system, server and front end and divides the image scheduling process of RSIMSS into three modules. The data retrieval module is responsible for retrieving remote sensing images from the PostgreSQL cluster via the Hilbert spatial index. The data transmission module is responsible for input/output of remote sensing image tiles from the HDFS file system according to the instructions of the user and administrator. The data visualization module is responsible for visualizing and rendering the remote sensing image tile data through the Leaflet, and passing the final view to the user layer for display. At the same time, three scheduling optimization strategies (ring cache mechanism, multi-threading mechanism and tile prefetching mechanism) for RSIMSS were also designed. According to the test results, the three optimization mechanisms work together to greatly improve the scheduling efficiency of image tiles, and realize real-time data scheduling and view roaming. Based on the spatial index algorithm and tile pyramid model, this paper advances the research on remote sensing image scheduling efficiency optimization from the perspective

of computer network transmission. The optimization strategy of RSIMSS provides a novel strategy for the real-time scheduling of remote sensing images.

However, there are still some areas for improvement in the RSIMSS process. First of all, due to hardware constraints, the small HDFS distributed file system deployed is limited to testing the I/O performance of RSIMSS and fails to test the capacity scalability and system stability of RSIMSS. In addition, the current RSIMSS mainly focuses on data storage, management and scheduling. Still, in the future, we hope to further develop remote sensing image processing functions for RSIMSS on the reserved function expansion interface so that RSIMSS covers the whole process of remote sensing images from retrieval and scheduling to application.

**Author Contributions:** Conceptualization, Zhen Zhang; methodology, Jiankun Zhu; software, Jiankun Zhu; validation, Jiankun Zhu and Zhen Zhang; formal analysis, Jiankun Zhu and Zhen Zhang; investigation, Jiankun Zhu; resources, Zhen Zhang and Jiankun Zhu; data curation, Jiankun Zhu; writing—original draft preparation, Jiankun Zhu; writing—review and editing, Jiankun Zhu, Zhen Zhang and Zhengnan Gu; Visualization, Jiankun Zhu, Zhen Zhang, Fei Zhao and Leilei Wang; supervision, Zhen Zhang, Fei Zhao and Haoran Su; project administration, Jiankun Zhu, Zhengnan Gu and Leilei Wang; funding acquisition, Zhen Zhang. All authors have read and agreed to the published version of the manuscript.

**Funding:** This research was funded by the Major Project on Natural Science Foundation of Universities in Anhui Province (Grant No. 2022AH040111) and the National Natural Science Foundation of China (Grant Nos. 42071085 and 41701087).

**Institutional Review Board Statement:** Not applicable.

**Informed Consent Statement:** Not applicable.

**Data Availability Statement:** The data presented in this study are available on request from the author.

**Acknowledgments:** We want to express our sincere gratitude to the anonymous reviewers and editors for their efforts to improve the paper.

**Conflicts of Interest:** The authors declare no conflict of interest.

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
