# Peer review of "Efficient Management and Scheduling of Massive Remote Sensing Image Datasets"

_ijgi, doi:10.3390/ijgi12050199_

Round 1

Reviewer 1 Report

My suggestion is that references should be in alfabet order.

Author Response

Response: Thank you for your suggestion, we have sorted out the references according to the format required by the journal.

Reviewer 2 Report

In this paper, the authors describe a remote sensing data management and scheduling system to store, retrieve, and visualize massive remote sensing data efficiently. Basically, this paper is well organized, but it needs to be revised to be clear, concise, and easy to understand with expected technical details. From my point of view, the authors should think about below:

1.      Improve many grammatical errors, misused words, or awkward phrases that lead to confusing or misunderstanding. It is suggested that a native English speaker should proofread this paper. Some are given here:

1)     Page 1, “such that a single image can contain gigabytes of information”

2)     Page 2, “The operation and monitoring of distributed storage need to be compensated for by technical means”

3)     Page 3, “The system also sets aside other functional expansion interfaces to facilitate future expansion of the system”

2.      The same information on the proposed strategy and approaches are repeated in sections of Methods, Results, and Conclusions. Please eliminate these redundancies;

3.      An extensible system architecture is proposed in Section 2.1. No technical details on how to add new functions in each layer are given;

4.      In Section 2.2.1, “its retrieval algorithm is perfect.” What “its” means here? Avoid using words like “prefect”. And this sentence is unclear and logically inconsistent here.

5.      Section 2.2.1 describes details on the well-known Hilbert space-filling curve. Describe it in a short and concise manner.

6.       In Section 2.2.2, if the metadata is not offered along with image data by the data sources like LandSat or MODIS data, how does the system obtain the required information? And how to calculate the image Hilbert code from its spatial information?

7.      In Table 2, why “double” is used for the “FileSize” field?

8.      Section 2.2.3 doesn’t give a clear picture of how NameNode and DataNodes work together for image data writing, query and reading. And Figure 6 is exactly same as the original figure (https://hadoop.apache.org/docs/r1.2.1/hdfs_design.html), source must be added.

9.      Too much details on building the pyramid layers are given. A short description is enough. Must the image size be N by N?  

10.    In Section 2.3.1, requesting image data of area of interest for the specified period (like LandSat-8 data for Africa from 2015 to 2022 to study drought) is a common requirement for remote sensing data management system, how does your system address such spatial and temporal requirements?

11.   In Section 2.3.3, how does Leaflet display the raster data in non-grid format (like HDF5, MODIS data mentioned in your experiment)?

12.   In Section 2.4.1, the ring caching mechanism in Figure 10 is not described clearly. Maybe using “rolling caching” is better.  “All write and read operations are carried out within a fixed storage space.” How much memory? How are the "read" and "write" pointers moving in the ring buffer?

13.   In Section 2.4.3, Figure 11, 12 and 13 are not necessary. A diagram of interactions between ring cache and multiple threads is needed to help understand the proposed mechanisms.  

14.   In Section 3.1.2, give a brief description on the experimental data (like source, spatial and temporal coverages, etc.).

15.   In Section 3.1.5, use other simple combinations of letters and symbols to represent the test cases;

16.   In Section 4, summarize the main points of the proposed methods in a few sentences, and discuss their applicability in high volumes of remote sensing data management system;

17.   Format the references in required format;

Author Response

In this paper, the authors describe a remote sensing data management and scheduling system to store, retrieve, and visualize massive remote sensing data efficiently. Basically, this paper is well organized, but it needs to be revised to be clear, concise, and easy to understand with expected technical details. From my point of view, the authors should think about below:

1. Improve many grammatical errors, misused words, or awkward phrases that lead to confusing or misunderstanding. It is suggested that a native English speaker should proofread this paper. Some are given here:

1) Page 1, “such that a single image can contain gigabytes of information”

2)Page 2, “The operation and monitoring of distributed storage need to be compensated for by technical means”

3)Page 3, “The system also sets aside other functional expansion interfaces to facilitate future expansion of the system”

Response: Thank you for pointing out the problems, I have modified them. In addition, we asked several high-quality native English editors to check and revise the manuscript to ensure the correctness of English language, grammar, punctuation, spelling and overall style.

2. The same information on the proposed strategy and approaches are repeated in sections of Methods, Results, and Conclusions. Please eliminate these redundancies;

Response: Thank you for your suggestions. For the repeated descriptions in the methods, results and conclusions, I have replaced them with simplified statements, eliminating some redundancy.

3. An extensible system architecture is proposed in Section 2.1. No technical details on how to add new functions in each layer are given;

Response: In order to facilitate the expansion of the system in the future, we have reserved data access, query and editing processing channels for subsequent development, and compiled them into interfaces in Java language. At present, RSIMSS mainly focuses on data storage, management and scheduling, but for the future, we hope to further expand more powerful data processing functions for RSIMSS.

4. In Section 2.2.1, “its retrieval algorithm is perfect.” What “its” means here? Avoid using words like “prefect”. And this sentence is unclear and logically inconsistent here.

Response: Here, “its” refers to the “basic data type” supported by the database mentioned above. In order to avoid ambiguity, I have modified this sentence.

5. Section 2.2.1 describes details on the well-known Hilbert space-filling curve. Describe it in a short and concise manner.

Response: Thanks for your advice, I have streamlined the description of the Hilbert space-filling curve.

6. In Section 2.2.2, if the metadata is not offered along with image data by the data sources like LandSat or MODIS data, how does the system obtain the required information? And how to calculate the image Hilbert code from its spatial information?

Response: Our data are downloaded from the existing remote sensing image data website, and the obtained data have a complete file format. Therefore, we all extract image information in the metadata file attached to the data source. We really didn't take into account the missing metadata file. However, the data source information of RSIMSS is stored in the "product set" table, which contains the average coverage of the image of the data source and the Hilbert coding level, so as long as the approximate center point coordinates of the image can be obtained, the Hilbert code of the image can also be calculated.

7. In Table 2, why “double” is used for the “FileSize” field?

Response: The file size of the image is hundreds of MB, and even reaches the GB level. If the file size of the image is represented by an integer type of bytes, the value will be too large, so we decide to represent the image file size in MB. There will inevitably be decimal places in MB, so the double type is more appropriate.

8. Section 2.2.3 doesn’t give a clear picture of how NameNode and DataNodes work together for image data writing, query and reading. And Figure 6 is exactly same as the original figure (https://hadoop.apache.org/docs/r1.2.1/hdfs_design.html), source must be added.

Response: Thank you for pointing out the problem, we have supplemented the description of HDFS storage, management and read data process in Section 2.2.3 to make it clearer. And add a picture source for Figure 6.

9. Too much details on building the pyramid layers are given. A short description is enough. Must the image size be N by N?

Response: Because remote sensing image tile pyramid is an important organization method of image unstructured data, and it is also one of the important ways to speed up the efficiency of image scheduling, this paper describes the process of remote sensing image tile pyramid slightly in detail. In the tile segmentation, the theory requires the number of remote sensing image pixels N * N, but in the actual construction of the image pyramid, it is difficult to ensure that the number of image pixels is the standard N * N. Therefore, when constructing the image pyramid, we use the long side of the image as N to perform N * N tile segmentation, and the short side of the insufficient N is complemented by blank pixels.

10. In Section 2.3.1, requesting image data of area of interest for the specified period (like LandSat-8 data for Africa from 2015 to 2022 to study drought) is a common requirement for remote sensing data management system, how does your system address such spatial and temporal requirements?

Response: RSIMSS can simultaneously support multi-conditional image retrieval based on time, space, data source and cloudiness. The time, data source and cloudiness of the image are stored in the PostgreSQL database table as the basic data type, while the spatial position can be converted into one-dimensional Hilbert coding after calculation. Hilbert coding is also stored in the PostgreSQL database table as an integer type. Finally, the multi-condition retrieval of remote sensing images can be realized by performing table record queries based on fields such as “Atime”, “GridCode”, “Cloud” and “PId” from the database in standard SQL language.

11. In Section 2.3.3, how does Leaflet display the raster data in non-grid format (like HDF5, MODIS data mentioned in your experiment)?

Response: RSIMSS does not directly display the original data of remote sensing images. All remote sensing images need to build an image tile pyramid. After tile segmentation and resampling, all image tiles are organized in GeoTIFF format and stored in HDFS. Leaflet is responsible for displaying image tile data in GeoTIFF format.

12. In Section 2.4.1, the ring caching mechanism in Figure 10 is not described clearly. Maybe using “rolling caching” is better.  “All write and read operations are carried out within a fixed storage space.” How much memory? How are the "read" and "write" pointers moving in the ring buffer?

Response: Thank you for pointing out the problem, our description of the ring cache mechanism is not detailed enough. In this regard, we supplement the description of ring cache in chapter 2.4.1. After tile segmentation and resampling of remote sensing images, the amount of data per tile is about 1~2MB. The main purpose of the ring cache is to act as a data transfer station for loading image tiles. Therefore, we set the size of each cache to 3MB to ensure that the cache can store image tiles normally. The movement of the read and write pointers of the ring cache depends on the multithreading mechanism. Therefore, we put the specific movement of the read and write pointers in the 2.4.3 chapter for detailed description, and add Figure 14 to explain the interaction process of the ring cache mechanism and the multithreading mechanism.

13. In Section 2.4.3, Figure 11, 12 and 13 are not necessary. A diagram of interactions between ring cache and multiple threads is needed to help understand the proposed mechanisms.

Response: According to your suggestion, we have added a diagram of the interaction between the ring caching mechanism and the multithreading mechanism in section 2.4.3. But for Figures 11, 12 and 13, we still think it would be more appropriate to keep them. Figures 11 and 12 can help the reader better understand the tile prefetching mechanism. Figure 13 provides a clearer illustration of the distributed deployment of PostgreSQL based on the Hilbert grid.

14. In Section 3.1.2, give a brief description on the experimental data (like source, spatial and temporal coverages, etc.).

Response: Thanks for your suggestion, we added a table in Section 3.1.2 to give a simple description of the experimental data.

15. In Section 3.1.5, use other simple combinations of letters and symbols to represent the test cases;

Response: Thank you for your suggestion, we use a combination of letters and symbols to represent test cases is indeed too complex, therefore, we have modified it to a more concise representation.

16. In Section 4, summarize the main points of the proposed methods in a few sentences, and discuss their applicability in high volumes of remote sensing data management system;

Response: Thank you for your suggestion, we have revised section 4, streamlined the description of the work done in this article, and added the following to section 4 : (1) Discussed the differences between our work and the current research status ; (2) Summarize our contributions in related fields ; (3) Some shortcomings and limitations of RSIMSS are discussed.

17. Format the references in required format;

Response: We have arranged the references according to the format required by the journal.

Reviewer 3 Report

Dear authors, My review comments are below, to help you restructure, revise, and improve the manuscript.

Comment 1: Please add a line number to the article!

Comment 2: More explanation is needed for where there is a research gap and what the goals of the research are. The research gap and the goals of the research are not explained in detail which leads to the reader missing the significance of the research.

Comment 3: Reference[10]-[16]: Please give the knowledge from your summarization rather than align a pile of literature.

Comment 4: The title of the article is "Efficient Management and Scheduling of Massive Remote Sensing Image Datasets", but the article does not mention the management of image data (unstructured data), only the processing and management of remote sensing metadata?

Comment5:As we all know, PostgreSQL has a PostGIS module that can manage remote sensing image data, and PostgreSQL can also be deployed in a distributed manner. Why do the authors use HDFS and PostgreSQL alone. Please explain it.

Comment6:In the references, some documents about MongoDB are also cited. In fact, MongoDB has good performance in managing image data, and it is also distributed. Why doesn't the author use MongoDB for distributed processing? Please explain it.

Comment7:The authors use the coordinates of the four corners in the metadata for spatial retrieval, which can indeed be realized in principle. However, have you compared the performance with spatial retrieval on image data? Please explain it.

Comment8:I rarely see an introduction to unstructured image data processing in articles, which will make readers feel that this article is just processing metadata, so I suggest changing the title to "Efficient Management and Scheduling of Massive Remote Sensing metadata".

Comment9:The titles of Figures 18 and 19 are too simple, please provide a brief explanation in the titles.

Comment10:Please make Figure 14 and Figure 15 more beautiful.

Comment11:Figure 16 should be added with other map elements such as a frame, scale, and compass.

Comment12:The authors must discuss the differences between their work and previous studies and the applicability of their findings/results and future study in this field.

Comment13: I would like to request the author to emphasis on the contributions on practically and academically in discussion session.

Author Response

Dear authors, My review comments are below, to help you restructure, revise, and improve the manuscript.

Comment 1: Please add a line number to the article!

Response: Thanks for your suggestion, we have added the line number to the manuscript.

Comment 2: More explanation is needed for where there is a research gap and what the goals of the research are. The research gap and the goals of the research are not explained in detail which leads to the reader missing the significance of the research.

Response: According to the problems you pointed out, we have supplemented the description of the current research status in the introduction, so that it can better reflect the difference between this paper and the current research, and make the research objectives of this paper more clear.

Comment 3: Reference[10]-[16]: Please give the knowledge from your summarization rather than align a pile of literature.

Response: For references [10] - [16], we not only list them, but also summarize and sort out the advantages and disadvantages of various distributed file systems and their applicable scenarios, and give a brief description later to explain why we choose HDFS. FastDFS, GridFS and GlusterFS are suitable for file-based online services, such as video and still images. Lustre requires the support of special devices typically used in high-performance computing. Remote sensing image files are very large—a single file can reach the gigabyte size—and once stored without modification, in line with the design concept of HDFS and MooseFS. However, MooseFS is typically applied to single-cluster deployments. As the cluster scale expands, it is prone to uneven loads, and a greater risk of instability.

Comment 4: The title of the article is "Efficient Management and Scheduling of Massive Remote Sensing Image Datasets", but the article does not mention the management of image data (unstructured data), only the processing and management of remote sensing metadata?

Response: In this paper, PostgreSQL database and HDFS distributed file system are used to manage the structured and unstructured data of remote sensing images in a hybrid manner. The remote sensing image metadata is managed by the PostgreSQL database, and the remote sensing image files and their image tile files are stored in HDFS. RSIMSS is not only to manage the metadata of remote sensing images, for image files, we first divide them into tiles, and then store the original files and tile data in HDFS, see Section 2.2.3 for details.

Comment5:As we all know, PostgreSQL has a PostGIS module that can manage remote sensing image data, and PostgreSQL can also be deployed in a distributed manner. Why do the authors use HDFS and PostgreSQL alone. Please explain it.

Response: Relational database is more suitable for the storage and management of structured data. It focuses on convenient data query and analysis capabilities and powerful transaction processing capabilities. Although the PostgreSQL database can also be deployed in a distributed manner and can be used to store remote sensing image data, this is limited to the limited amount of data. When the amount of data is massive, the PostgreSQL database cluster has high expansion cost, high cluster maintenance cost, and low throughput of data writing and reading. The cost performance of storing massive unstructured image files is extremely low, and the advantages of relational databases cannot be fully utilized. The distributed file system HDFS has high data throughput, high concurrency and strong capacity scalability, and the capacity expansion cost is low, which has inherent advantages in storing massive large files. But at the same time, the distributed file system does not have the powerful data query and analysis ability of relational database, which is necessary for image retrieval. Therefore, this paper combines PostgreSQL relational database with HDFS distributed file system, uses PostgreSQL database to manage structured metadata of remote sensing images, and uses HDFS to store unstructured file data of remote sensing images, giving full play to the advantages of both. It not only realizes efficient management and query of images with relational database, but also solves the problem of massive image file storage with HDFS.

Comment6:In the references, some documents about MongoDB are also cited. In fact, MongoDB has good performance in managing image data, and it is also distributed. Why doesn't the author use MongoDB for distributed processing? Please explain it.

Response: In fact, we have also considered using MongoDB, but we finally chose to use PostgreSQL database to manage remote sensing image metadata for two reasons:

(1) PostgreSQL includes an extension module called PostGIS, which supports the storage and management of geographic objects and integrates the most comprehensive geographic processing functions. These geographic processing functions, especially the geospatial measurement function, can provide great help for the spatial location retrieval of remote sensing images. It can help us quickly calculate the location relationship between the retrieval area and the Hilbert grid, so as to realize the rapid positioning of the retrieval area.

(2) We refer to some related literature and compare PostgreSQL with MongoDB. The final results show that PostgreSQL is better than MongoDB in distributed deployment. These documents are also cited in the article:

[27]Zhou, X.; Wang, X.; Zhou, Y.; Lin, Q.; Zhao, J.; Meng, X. RSIMS: Large-Scale Heterogeneous Remote Sensing Images Management System. Remote Sensing 2021, 13, 1815, doi:10.3390/RS13091815.

[30]Makris, A.; Tserpes, K.; Spiliopoulos, G.; Zissis, D.; Anagnostopoulos, D. MongoDB Vs PostgreSQL: A comparative study on performance aspects. GeoInformatica 2020, 1-25, doi:10.1007/s10707-020-00407-w.

Comment7:The authors use the coordinates of the four corners in the metadata for spatial retrieval, which can indeed be realized in principle. However, have you compared the performance with spatial retrieval on image data? Please explain it.

Response: We do not use the coordinates of the four corners of the image in the metadata for spatial indexing, and there is no such description in the manuscript. We divide the global region into multi-level Hilbert grids, and then calculate the Hilbert code corresponding to the image with the center point coordinates of the image and the coverage of the image, and use the code to construct a one-dimensional spatial index of the remote sensing image metadata. When performing image retrieval, we also iteratively divide the retrieval area into multi-level Hilbert grids, and calculate the Hilbert coding set corresponding to the retrieval area, so that when querying from the PostgreSQL database, we only need to query the remote sensing image data within the coding set, as described in Chapter 2.3.1. Finally, in chapter 3.1.4, we also compare the designed multi-level Hilbert grid retrieval algorithm with the spatial index algorithm of PostgreSQL database, and the results show that the multi-level Hilbert grid spatial index has higher retrieval performance.

Comment8:I rarely see an introduction to unstructured image data processing in articles, which will make readers feel that this article is just processing metadata, so I suggest changing the title to "Efficient Management and Scheduling of Massive Remote Sensing metadata".

Response: Thank you for your suggestions. Maybe we focus more on the tile segmentation and storage of the image in Chapter 2.2.3. There is too little description of the storage of the original image file, so you feel that we do not store and manage unstructured image data. In fact, for the unstructured image file of the image, we store it and the segmented tile data in the HDFS distributed file system. In this regard, we supplement the description of the storage of the original image file in chapter 2.2.3, so that readers can better understand the storage of unstructured image files in this paper.

Comment9:The titles of Figures 18 and 19 are too simple, please provide a brief explanation in the titles.

Response: Thanks for your suggestion, we have supplemented the titles of Figure 18 and Figure 19 so that they can accurately express the meaning in the figure.

Comment10:Please make Figure 14 and Figure 15 more beautiful.

Response: Thanks for your suggestion, we have improved Figure 14 and Figure 15 to make it more beautiful.

Comment11:Figure 16 should be added with other map elements such as a frame, scale, and compass.

Response: Thank you for your advice. The purpose of Figure 16 is to show the three retrieval areas we used to test the retrieval performance of RSIMSS. It is intercepted from RSIMSS rather than drawn with professional mapping software. Therefore, it is difficult to add other map elements such as scale and compass to Figure 16.

Comment12:The authors must discuss the differences between their work and previous studies and the applicability of their findings/results and future study in this field.

Response: Thank you for your suggestion, we have revised Chapter 4. In Chapter 4, we discuss the differences between our work and the current research status, and discuss some shortcomings and limitations of our current work.

Comment13: I would like to request the author to emphasis on the contributions on practically and academically in discussion session.

Response: Thank you for your suggestions. In Chapter 4, we add a description of the contribution of our work to the current research status in related fields.

Round 2

Reviewer 2 Report

The authors have addressed all my concerns. Thanks!

It is recommended that the paper should be accepted for publication after text editing.

Author Response

Thank you for your contribution to our manuscript review, your comments and suggestions have greatly helped us improve our manuscript. We revised the manuscript based on your suggestions and again asked a native English editor to check the manuscript to ensure that the English language, grammar, punctuation, spelling, and overall style of the manuscript were correct.

Reviewer 3 Report

The quality of the article has been improved to a certain extent through revision, but there are still some problems. The authors still fails to let readers clearly see the advanced nature of the proposed method, and does not describe how to deal with remote sensing raster data in detail. At least I think that according to your title, this part of the content must be clearly understood by readers. The processing method of metadata is very mature now, and the processing of raster data is the most important.

Therefore, it is recommended to make improvements in the following aspects:

(1) Add content about the specific processing of raster data, instead of replacing it with the processing flow of HDFS;
(2) Standardize the presentation of some pictures in the article;
(3) Clarify the advancement and advantages of the method proposed in the article.

Author Response

Thank you for reviewing our manuscripts and making valuable comments. We carefully considered your questions and suggestions and made further improvements to the manuscript based on your comments. The following is our reply to the review comments.

Comment 1: Add content about the specific processing of raster data, instead of replacing it with the processing flow of HDFS;

Response: Thank you very much for your suggestion. Regarding the raster data of remote sensing images, except for the construction of image tile pyramids for them, we have not processed them too much. The focus of this paper is to achieve efficient storage, management and scheduling of massive remote sensing images. Therefore, for massive raster data, we do not need to process the original files of remote sensing images too much. We only need to use the HDFS file system to store the original files and image tiles of massive remote sensing images. The packaging of image tile data and the storage process of remote sensing images are realized by the JAVA class provided by Hadoop for HDFS, so we describe the storage process of HDFS in detail.

Comment 2: Standardize the presentation of some pictures in the article;

Response: Thank you for your suggestion, we checked the format of all the pictures in the manuscript to ensure that the format is correct.

Comment 3: Clarify the advancement and advantages of the method proposed in the article.

Response: Thank you very much for your comments. We have revised the description of the advancement and advantages of the paper in the introduction and discussion to make it clearer. The advanced nature and advantages of the methods used in this paper are as follows :

(1) Scheduling remote sensing images is more computer-oriented research, and its process relies on computer hardware and network transmission. Presently, most of the research on remote sensing image scheduling efficiency optimization methods stays in the spatial index algorithm and tile pyramid model, which have not been further deepened. Based on the research of spatial index algorithm and tile pyramid model, this paper deeply analyzes the scheduling process of remote sensing image from the perspective of computer network transmission, and designs three new scheduling mechanisms (ring caching, multi-threading and tile-prefetching mechanism) to comprehensively optimize the scheduling process. The three mechanisms cover the whole process of computer network transmission of image tiles between storage system, cache and front end. Through the three mechanisms to work together, the remote sensing image data scheduling to achieve second-level real-time response.

(2) According to the spatial distribution characteristics of massive multi-source heterogeneous remote sensing image datasets, a spatial index based on a multi-layer Hilbert grid is constructed to achieve efficient retrieval. The PostgreSQL database cluster is distributed with the same Hilbert grid, and the multi-threading mechanism is relied on in the cluster to greatly improve the efficiency of cross-server and cross-database image retrieval. The multi-level Hilbert grid division method enables the system to quickly filter out a large number of images when performing spatial queries and achieve accurate retrieval area positioning. Thanks to the extremely high locality retention of the Hilbert curve, deploying the PostgreSQL cluster according to the Hilbert grid division method avoids cross-server data retrieval as much as possible. Even when this cannot be avoided, retrieval efficiency can be improved by using multi-threaded parallel retrieval.

(3) Distributed file systems have gradually emerged due to their powerful capacity expansion capabilities and data throughput, becoming one of the mainstream ways to store massive remote-sensing images. However, the data retrieval method of the distributed file system is based on the file, and it is difficult to identify the specific attribute information inside the file. In this Paper, the distributed file system and relational data library are used to manage remote sensing image data in a hybrid manner, HDFS with high I/O performance and high capacity scalability is used to store massive unstructured file data, and PostgreSQL relational database with powerful retrieval performance is used to manage structured metadata. RSIMSS takes full advantage of the respective advantages of distributed file systems and relational databases. At the same time, the system realizes fully automated image tile pyramid construction and image metadata analysis.